# De novo DNA methyltransferase activity in colorectal cancer is directed towards H3K36me3 marked CpG islands

Roza H. Ali Masalmeh[1], Francesca Taglini[1,2], Cristina Rubio-Ramon[1,2], Kamila I. Musialik[1,2], Jonathan Higham[1], Hazel Davidson-Smith[1], Ioannis Kafetzopoulos [1,2], Kamila P. Pawlicka[1,2], Hannah M. Finan[1,2], Richard Clark [3], Jimi Wills[2], Andrew J. Finch[2,4], Lee Murphy [3] & Duncan Sproul [1,2✉]

The aberrant gain of DNA methylation at CpG islands is frequently observed in colorectal tumours and may silence the expression of tumour suppressors such as *MLH1*. Current models propose that these CpG islands are targeted by de novo DNA methyltransferases in a sequence-specific manner, but this has not been tested. Using ectopically integrated CpG islands, here we find that aberrantly methylated CpG islands are subject to low levels of de novo DNA methylation activity in colorectal cancer cells. By delineating DNA methyltransferase targets, we find that instead de novo DNA methylation activity is targeted primarily to CpG islands marked by the histone modification H3K36me3, a mark associated with transcriptional elongation. These H3K36me3 marked CpG islands are heavily methylated in colorectal tumours and the normal colon suggesting that de novo DNA methyltransferase activity at CpG islands in colorectal cancer is focused on similar targets to normal tissues and not greatly remodelled by tumourigenesis.

[1] MRC Human Genetics Unit, IGMM, University of Edinburgh, Edinburgh, UK. [2] CRUK Edinburgh Centre, IGMM, University of Edinburgh, Edinburgh, UK. [3] Edinburgh Clinical Research Facility, University of Edinburgh, Edinburgh, UK. [4] Present address: Centre for Tumour Biology, Barts Cancer Institute, Queen Mary University of London, London, UK. ✉email: Duncan.Sproul@igmm.ed.ac.uk

DNA methylation is an epigenetic mark associated with gene repression. It is normally pervasive in mammalian genomes but absent from many regulatory elements, particularly CpG islands (CGIs)[1]. In tumours, CGIs often become aberrantly methylated. In some cases hypermethylated CGIs correspond to the promoters of tumour suppressor genes such as *MLH1, CDKN2A (p16/ARF)* and *BRCA1*[2]. At these genes, hypermethylation associates with repression and thus could drive tumourigenesis. In support of this hypothesis, targeted methylation of the *CDKN2A* promoter in mammary epithelial cells prevents their entry into senescence[3]. However, aberrant CGI hypermethylation also occurs at many other CGIs that are not obviously the promoters of tumour suppressor genes[2]. These aberrantly methylated CGIs are often repressed by polycomb repressive complexes and marked by H3K27me3 in the normal cells that give rise to the cancer[4–6]. However, the mechanisms underpinning the aberrant hypermethylation of CGIs in cancer remain unclear.

DNA methylation is established and maintained in human cells by the DNA methyltransferases (DNMTs): DNMT1, DNMT3A and DNMT3B[7]. DNMT3A and DNMT3B are de novo methyltransferases that establish methylation patterns during early development[8]. DNMT1 is the main enzyme responsible for maintaining DNA methylation[9]. It has a preference for acting at hemi-methylated sites in vitro[10] and also interacts with PCNA at replication forks[11].

DNMT3B is the methyltransferase most often implicated in the aberrant methylation of CGIs in cancer. DNMT3B levels are often increased in cancers relative to normal tissues and higher levels in colorectal tumours correlate with the aberrant methylation of several CGIs[12,13]. Aberrant methylation at CGIs in cancer could be programmed by the sequence-specific recruitment of DNMTs through transcription factors[14]. The transcription factor MAFG is proposed to directly recruit DNMT3B to CGIs methylated in BRAF mutant colorectal tumours[15]. A parallel study suggested that in KRAS mutant colorectal cancer, DNMT1 is recruited by ZNF304 to cause de novo methylation of several genes including *CDKN2A*[16]. However, in mouse embryonic stem (ES) cells, DNMT3B is targeted to regions marked by H3K36me3[17]. These observations suggest a model in which DNMT activity in cancer cells is remodelled to target H3K27me3 marked CGIs in a sequence-specific manner resulting in their aberrant methylation. Despite this, previous studies have not measured de novo DNMT activity at CGIs in cancer cells and it is unclear whether it is indeed elevated at those that are aberrantly methylated.

Here we use different experimental strategies to assay de novo DNMT activity at CGIs in colorectal cancer cells. In contrast to current models, we find that the highest levels of de novo DNMT activity are found at CGIs that are marked by H3K36me3 and methylated in the normal colon.

## Results

**CGIs are not de novo methylated at ectopic locations in colorectal cancer cells.** The ectopic integration of CGIs has been used to demonstrate strong sequence-specific programming of CGI DNA methylation levels by transcription factors in mouse ES cells[18]. Therefore, in order to understand whether de novo DNMTs are targeted to CGIs that are aberrantly methylated in colorectal cancer in a sequence-specific manner we asked whether these CGIs become methylated when integrated into ectopic locations in the genome of colorectal cancer cells. If DNMTs were specifically recruited to these CGIs in a sequence-specific manner, the initially unmethylated ectopic integrants would be expected to rapidly acquire high levels of DNA methylation.

We used piggyBac transposons to randomly integrate copies of 10 CGIs into the genome of HCT116 cells (Fig. 1a). We tested 6

CGIs that are frequently aberrantly methylated in clinical colorectal tumours (Supplementary Fig. 1a) and methylated in HCT116 cells (*CDKN2A, SFRP1, ZFP42, GATA4, EPHB1, CDH7* and *CDH13*), a housekeeping CGI promoter that does not become hypermethylated in colorectal cancer, *BUB1*, and a normally methylated CGI, *DAZL* (Supplementary Fig. 1a). The CGI promoter of the tumour suppressor gene *MLH1* that can be methylated in colorectal cancer[19,20] (Supplementary Fig. 1a) but is unmethylated in HCT116 cells was also tested. Cell populations carrying CGI integrations were then expanded for 4 weeks to ensure free plasmid was no longer present in the population before we determined DNA methylation levels at the integrated copies and native loci using specific bisulfite PCR primers. Each ectopically integrated copy assayed will derive from a separate integration site in the population of transfected cells thus sampling a diverse array of different genomic locations.

The *BUB1* and *MLH1* CGIs both remained unmethylated when integrated into ectopic locations using piggyBac (Fig. 1b). The vast majority of integrated copies of the aberrantly methylated CGIs tested also did not become de novo methylated (Fig. 1b, c). Rare cases of methylation observed at integrated copies were low level and heterogeneous (Fig. 1c) but confirmed that HCT116 cells are capable of de novo methylation as previously reported[21]. The highest levels of methylation observed were at ectopic copies of the *CDH13* CGI but these were still much lower than those seen at the native locus (Fig. 1b). Ectopic copies of the *DAZL* CGI also did not become methylated to a high level (Fig. 1b) suggesting that it is only targeted by de novo DNMTs when it gains methylation early in development[22]. In order to confirm that this result was not specific to HCT116 cells, we repeated the experiment in another colorectal cancer cell line, RKO cells. We tested 4 CGIs, *MLH1, CDKN2A, SFRP1* and *CDH7* that are all methylated at their native locations in RKO cells. Ectopically integrated copies of these CGIs were mostly unmethylated in RKO cells (Supplementary Fig. 1b, c).

Overall, the results of this experiment suggest that in colorectal cancer cells surprisingly low de novo DNMT activity is targeted in a sequence-specific manner to those CGIs that frequently become hypermethylated in colorectal cancer.

**DNMT3B methylates H3K36me3 marked CGIs in colorectal cancer cells.** Given that our experiments with ectopically integrated CGIs suggested that aberrantly methylated CGIs were not strongly targeted by de novo DNMT activity, we next sought to determine which CGIs were targeted by DNMTs in colorectal cancer. We focused on DNMT3B because of its prior associations with aberrant methylation of CGIs in colorectal cancer[12,13]. We therefore reintroduced DNMT3B2, the major catalytically active DNMT3B isoform expressed in somatic cells (henceforth referred to as DNMT3B)[23], into hypomethylated HCT116 cells which lack DNMT3B and express low levels of a truncated DNMT1 product (DKO cells)[24,25]. Regions of the genome targeted by DNMT3B will gain methylation in this experiment. DNMT3B expression in DKO cells led to increased total methylation levels, as measured by mass-spectrometry, from 60.6 to 81.1% of the level observed in HCT116 cells (gain of 20.5%, Supplementary Fig. 2a). A gain of 9.04% was also observed when catalytically dead DNMT3B was reintroduced (Supplementary Fig. 2a).

We then used reduced representation bisulfite sequencing (RRBS)[26] to determine which CGIs gained methylation in DKO cells upon reintroduction of DNMT3B. In this experiment 2238 CGIs gained significant levels of methylation and are putative DNMT3B targets (Fig. 2a, ≥20% methylation gain and Benjamini–Hochberg corrected Fisher's exact tests $p < 0.05$). These CGIs were significantly enriched in a number of GO-

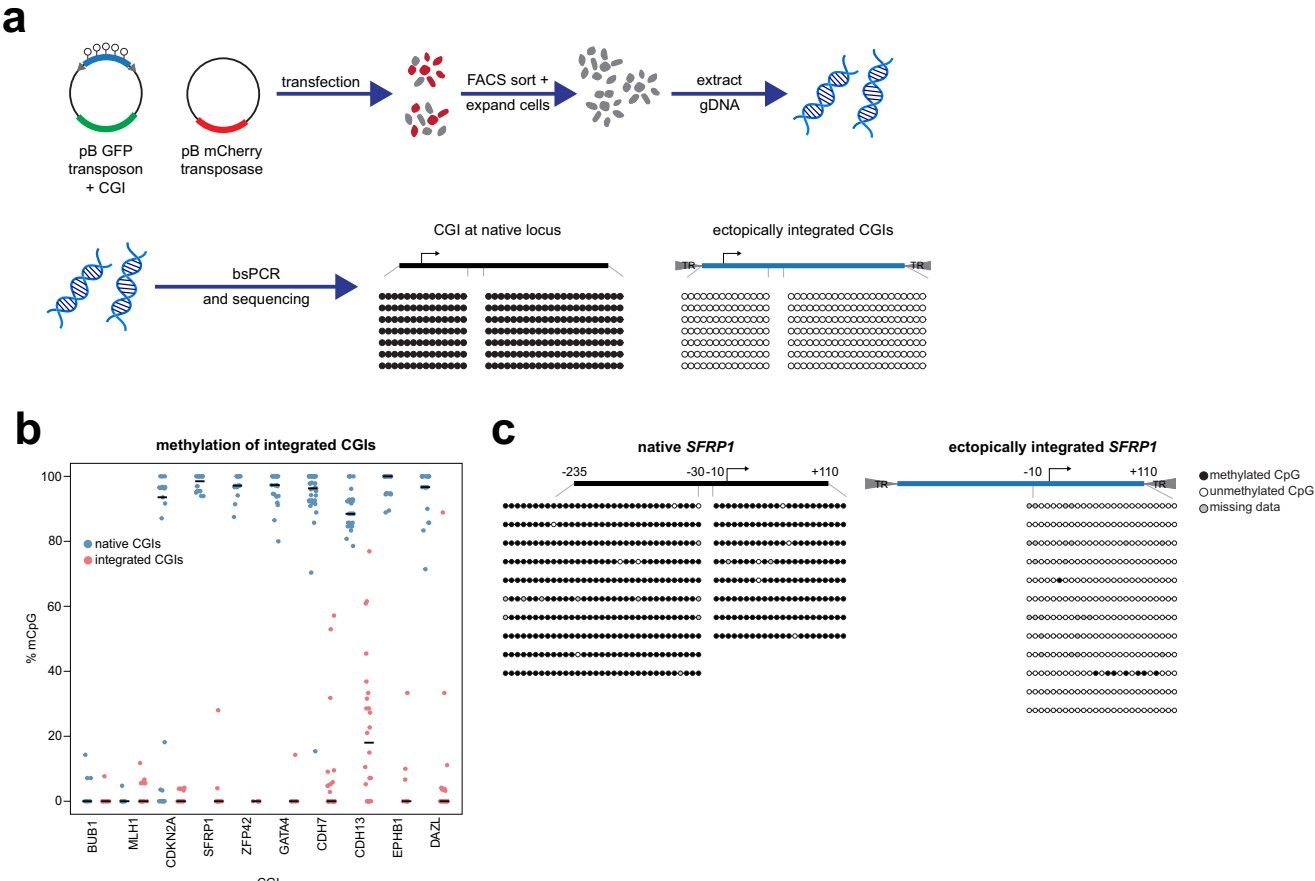

**Fig. 1 CGIs are not de novo methylated at ectopic locations in colorectal cancer. a** Schematic of experimental approach. Cells are transfected with piggyBac (pB) transposons containing unmethylated CGI DNA sequences along with the pB transposase. Transfected cells are selected by FACS based on fluorescent markers encoded by the plasmids and expanded to dilute out unintegrated copies of the plasmids. Genomic DNA is then extracted and subject to bisulfite PCR (bsPCR) to assay DNA methylation levels at the native and integrated copies of the CGI. **b** Ectopically integrated aberrantly methylated CGIs do not become de novo methylated in HCT116 cells. Plot showing the mean methylation per clone as assayed by bsPCR for 10 CGIs in their native location and when ectopically integrated in a piggyBac transposon. Each point represents the mean methylation level for a single bsPCR clone. The thick line indicates the median for each CGI. The number of clones analysed per CGI is indicated in Supplementary Table 1. **c** SFRP1 is not hypermethylated when integrated into ectopic locations in HCT116 cells. Illustrative example bsPCR data from panel **b** showing the SFRP1 promoter CGI in its native and integrated state. Circles are CpGs with different clones arranged vertically. Each integrated clone derives from a separate genomic integration. Black circles are methylated CpGs and white circles are unmethylated CpGs. Grey circles represent missing data due to sequencing errors. Source data are provided as a Source Data file.

terms but the enrichments we observed were very low (Supplementary Fig 2b, full list Supplementary Table 3). However, when cross-referenced to HCT116 histone modification ChIP-seq data from ENCODE[27], these CGIs were significantly enriched in regions marked by H3K36me3 (Fig. 2b, c). To directly examine the relationship between H3K36me3 and gain of methylation upon DNMT3B expression, we ranked CGIs by their H3K36me3 level in DKO cells measured by ChIP-Rx-seq[28]. H3K36me3 levels at CGIs in DKO cells were highly correlated with those in HCT116 cells (Spearman's Rho = 0.610, $p < 2.2 \times 10^{-16}$, Supplementary Fig. 2c). The gain in DNA methylation observed at CGIs when DNMT3B was introduced into DKO cells was significantly correlated to their level of H3K36me3 (Spearman's Rho = 0.584, $p < 2.2 \times 10^{-16}$, Fig. 2d). H3K36me3 is primarily observed in the bodies of actively transcribed genes[29]. Consistent with this observation, our putative DNMT3B target CGIs were significantly enriched in CGIs located in the bodies of coding genes (64.08%, $p < 2.2 \times 10^{-16}$ by one-sided Fisher's exact test versus all CGIs, Supplementary Fig. 2d).

We then conducted a second experiment, expressing DNMT3B to a higher level (using the CAG promoter rather than EF-1α,

Supplementary Fig. 2e). This resulted in a greater gain of methylation at H3K36me3 marked CGIs and more CGIs were restored to the levels observed in HCT116 cells (Supplementary Fig. 2f, g). Ectopic gains of methylation at loci hypomethylated in HCT116 cells were also observed in this 2nd experiment (Supplementary Fig. 2f, arrow).

Catalytically dead DNMT3B also caused a significant gain in DNA methylation at H3K36me3 marked CGIs compared to a GFP expressing control, although this was to a lower level than that seen with catalytically active DNMT3B (Supplementary Fig. 2f, g). The level of methylation gain seen with catalytically dead DNMT3B was similar in both the low and high expression experiments (Supplementary Fig. 2g). DNMT3A and DNMT3B are known to interact[30]. Interactions between catalytically dead DNMT3B and DNMT3A could therefore be responsible for these gains as DNMT3A is upregulated in DKO cells compared to HCT116 cells (Supplementary Fig. 2h). To further investigate the role of DNMT3A we overexpressed the somatic isoform, DNMT3A1, in DKO cells (henceforth referred to as DNMT3A)[31]. This caused gains of DNA methylation at H3K36me3 marked CGIs (Supplementary Fig. 2i) but these were less than that seen with DNMT3B

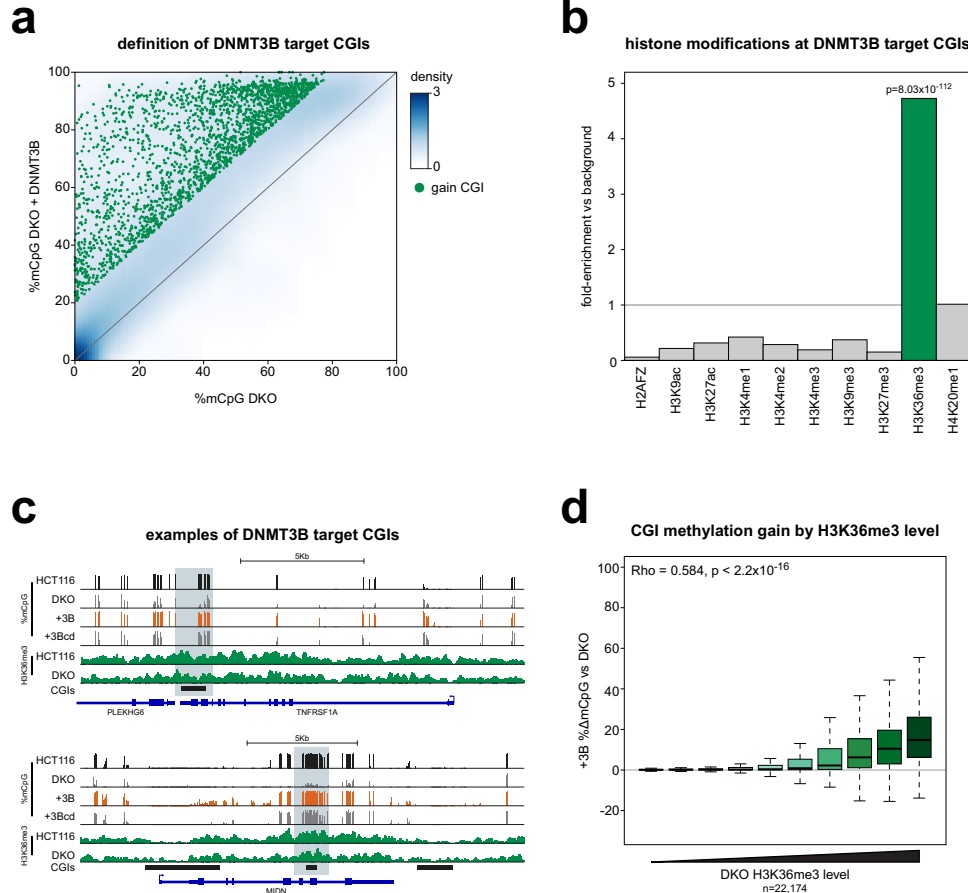

**Fig. 2 DNMT3B methylates H3K36me3 marked CGIs in colorectal cancer cells. a** Many CGIs gain methylation upon expression of DNMT3B in DKO cells. Density scatter plot comparing mean CGI methylation levels in DKO cells and DKO cells expressing DNMT3B (DKO + DNMT3B). Green individual points denote CGIs significantly gaining methylation upon DNMT3B expression (≥20% methylation gain and Benjamini–Hochberg corrected two-sided Fisher's exact tests $p < 0.05$). **b** DNMT3B target CGIs are enriched in H3K36me3 marked regions. Barplot of the fold-enrichment observed for HCT116 histone modification peaks in the set of CGIs that significantly gain methylation when DNMT3B is expressed in DKO cells compared to the background of all CGIs observed in the experiment. *P*-values are from Fisher's exact tests (1-sided for enrichment). **c** Examples of DNMT3B target CGIs. Genome browser plots showing DNA methylation levels with HCT116 and DKO H3K36me3 ChIP signal. CGIs and genes are shown below the plots. CGIs gaining methylation when DNMT3B is expressed in DKO cells are indicated in light blue. +3B = DKO + DNMT3B; +3Bcd = DKO + catalytically dead DNMT3B. Scale for methylation data is 0 to 100%. For H3K36me3 it is 0 to 0.24 normalised reads per 10⁶. **d** CGIs with the highest levels of H3K36me3 gain the greatest levels of DNA methylation. Boxplot showing gain of DNA methylation at CGIs when DNMT3B is expressed in DKO cells relative to their level of H3K36me3. CGIs were ranked by H3K36me3 level in DKO cells estimated from 2 biological replicates and split into 10 equally sized groups. Lines = median; box = 25th–75th percentile; whiskers = 1.5 × interquartile range from box. Source data are provided as a Source Data file.

(Fig. 2c). This suggests DNMT3A can also target H3K36me3 marked CGIs, albeit inefficiently. We then tested whether DNMT3B can recruit DNMT3A to H3K36me3 marked CGIs by performing ChIP for DNMT3A from HCT116 cells and HCT116 cells lacking DNMT3B (DNMT3B KO cells) where T7-tagged DNMT3A was expressed to equivalent levels (Supplementary Fig 2j). T7-DNMT3A was recruited to each of the H3K36me3 marked CGIs examined in DNMT3B KO cells, but it was recruited to a significantly greater extent in HCT116 cells suggesting DNMT3B can facilitate the recruitment of DNMT3A to these loci (Supplementary Fig. 2k).

Taken together these experiments suggest that DNMT3B activity is highest at CGIs that are marked by H3K36me3 in colorectal cancer. They also suggest that DNMT3A can be recruited by DNMT3B to H3K36me3 marked CGIs.

**DNMT3B is recruited to H3K36me3 marked CGIs**. To confirm that DNMT3B normally targets CGIs gaining methylation in our DKO experiments, we used CRISPR to introduce an N-terminal

T7 tag on DNMT3B in HCT116 cells (T7-DNMT3B cells) and performed ChIP-qPCR. N-terminal tagged DNMT3B is catalytically active in vivo[32] and we observed no loss of methylation at representative CGIs in T7-DNMT3B cells (Supplementary Fig. 3a). DNMT3B was significantly enriched at gene body CGIs in the *VWA1* and *TNFRSF1A* genes that are methylated in HCT116 and gained methylation when DNMT3B was expressed in DKO cells (Supplementary Fig. 3b). No enrichment of DNMT3B was seen at the *BRCA2* promoter CGI, a housekeeping gene that does not gain methylation in DKO cells expressing DNMT3B (Supplementary Fig. 3b).

We then performed T7-DNMT3B ChIP-Rx-seq to examine the determinants of DNMT3B localisation at CGIs more generally. DNMT3B target CGIs uncovered in DKO cells were enriched in DNMT3B binding versus other CGIs ($p$-value $< 2.2 \times 10^{-16}$, Wilcoxon rank-sum test, Fig. 3a, b). An analysis of all available HCT116 histone modification ChIP-seq data from ENCODE demonstrated that DNMT3B levels at CGIs are more strongly predicted by H3K36me3 levels than other histone marks (Fig. 3c).

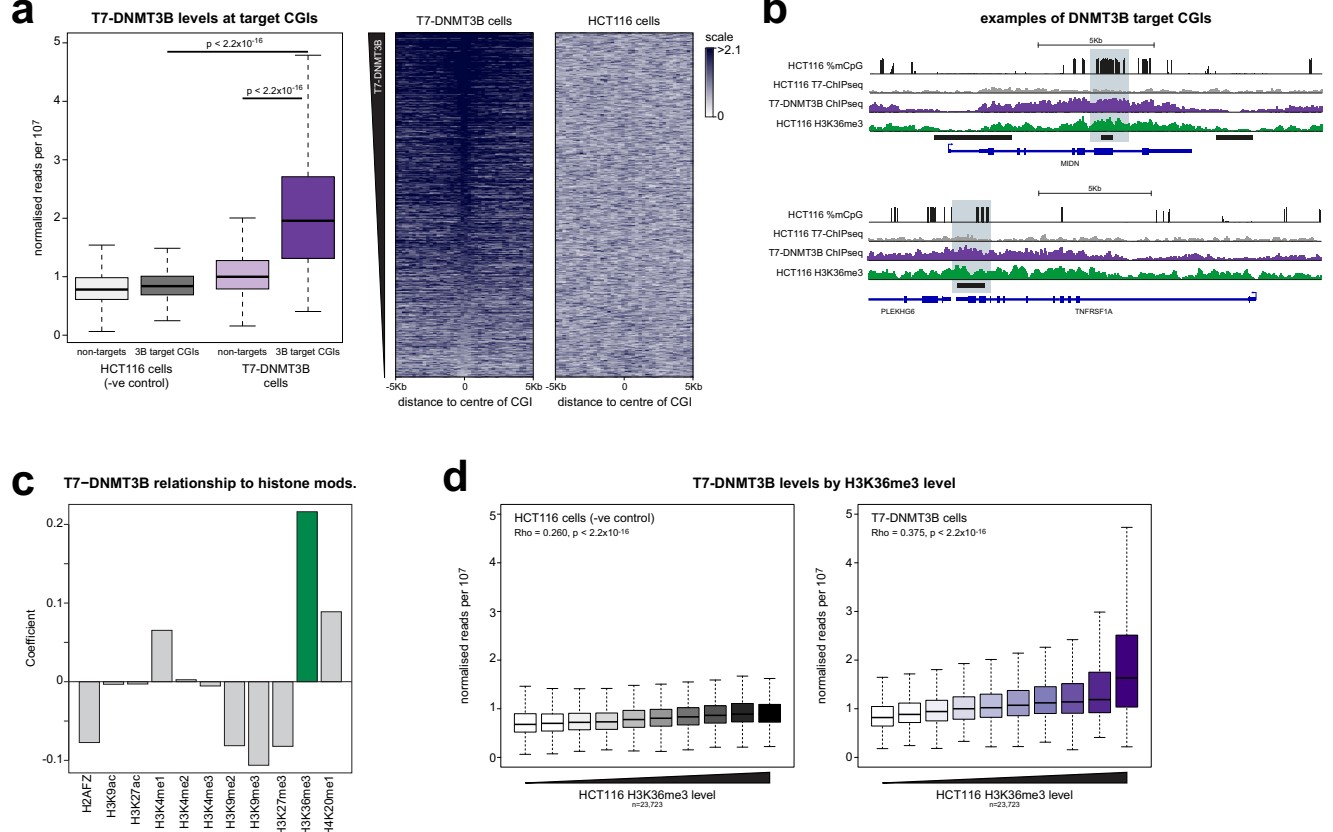

**Fig. 3 DNMT3B is recruited to H3K36me3 marked CGIs. a** DNMT3B localises to CGIs that gain methylation upon DNMT3B expression in DKO cells. Left, boxplot comparing levels of DNMT3B at putative DNMT3B targets ($n = 2238$) and other CGIs ($n = 19,939$). Right, heatmap showing measured DNMT3B around putative DNMT3B targets compared to the background in HCT116 cells where DNMT3B is untagged. Both panels show the mean of two biological replicates. Lines = median; box = 25th–75th percentile; whiskers = 1.5 × interquartile range from box. *P*-values are from two-sided Wilcoxon rank sum tests. **b** Examples of DNMT3B target CGIs showing DNMT3B levels. Genome browser plots showing DNA methylation levels and H3K36me3 ChIP signal in HCT116 along with T7-DNMT3B ChIP-seq. CGIs and genes are shown below the plots. DNMT3B target CGIs are indicated in light blue. Scale for methylation data is 0 to 100%. For DNMT3B signal it is 0 to 0.31 and for H3K36me3 it is 0 to 0.24 normalised reads per $10^6$. **c** DNMT3B levels at CGIs are most strongly related to H3K36me3 levels. Barplot showing the coefficients estimated in a linear model predicting DNMT3B levels at CGIs from all available histone modification ChIP-seq data from ENCODE. **d** CGIs with the highest levels of H3K36me3 show the highest levels of DNMT3B. Boxplot showing DNMT3B levels at CGIs relative to their level of H3K36me3 along with the levels of background in HCT116 cells where DNMT3B is untagged. CGIs were ranked by H3K36me3 level in HCT116 cells and split into 10 equally sized groups. Both panels show the mean of two biological replicates. Lines = median; box = 25th–75th percentile; whiskers = 1.5× interquartile range from box. *P*-values are from Spearman's correlation tests. Source data are provided as a Source Data file.

DNMT3B levels were also significantly correlated with H3K36me3 levels in our own HCT116 ChIP-Rx-seq (Fig. 3d, Spearman correlation, $p < 2.2 \times 10^{-16}$).

To confirm the localisation of DNMT3B in colorectal cancer cells, we then assayed the localisation of ectopically expressed T7-DNMT3B in RKO cells using ChIP-qPCR. A selection of DNMT3B target CGIs from DKO cells were marked by H3K36me3 in RKO cells (Supplementary Fig. 3c). DNMT3B was strongly localised to 3 of them (*TNFRSF1A*, *MIDN*, *MAP1S*) and detected above background at *VWA1* (Supplementary Fig. 3d). A negative control, *BRCA2*, lacked both H3K36me3 and DNMT3B (Supplementary Fig. 3d).

Taken together, these experiments confirm that the CGIs marked by H3K36me3 are those which are most strongly targeted by DNMT3B in colorectal cancer cells.

**H3K36me3 marked CGIs preferentially recover methylation following pharmacological hypomethylation**. To quantify de novo DNMT activity at CGIs without ectopically overexpressing

or focusing on specific DNMTs, we hypomethylated HCT116 cells using the demethylating drug 5-aza-2'-deoxycytidine (5-aza-dC) before measuring their recovery of DNA methylation. At 3 days following 5-aza-dC treatment, total DNA methylation had decreased to 49.9% of that in untreated cells (Supplementary Fig. 4a). HCT116 cells recovered methylation to levels similar to untreated cells within 22 days after 5-aza-dC treatment (Supplementary Fig. 4a). In order to compare the relative recovery rate of different CGIs, we performed RRBS across this time-course (Fig. 4a). The re-methylation observed in this experiment could be the result of de novo DNMT activity or the outgrowth of cells that escaped 5-aza-dC induced hypomethylation. However, we observed significant heterogeneity in the normalised recovery rates for different CGIs ($p = 1.31 \times 10^{-25}$ by ANOVA, see materials and methods) suggesting that the recovery of methylation was not solely explained by the outgrowth of cells escaping 5-aza-dC-induced hypomethylation.

H3K36me3 marked CGIs lost significantly more methylation than other HCT116 methylated CGIs at 3 and 6 days following 5-aza-dC treatment (Fig. 4b). However, we observed that

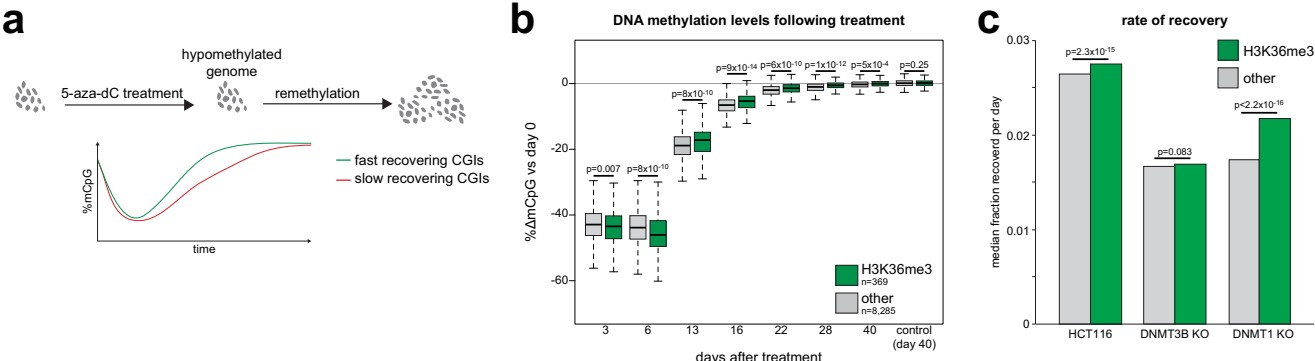

**Fig. 4 H3K36me3 marked CGIs preferentially recover methylation following pharmacological hypomethylation. a** Schematic of experimental approach. 5-aza-dC is used to hypomethylate cells and the kinetics of methylation recovery are compared for different CGIs. **b** H3K36me3 marked CGIs preferentially recover methylation following 5-aza-dC treatment. Boxplots of relative methylation at H3K36me3 marked CGIs ($n = 369$) and all other CGIs methylated ($n = 8285$) in HCT116 cells. *P*-values are from two-sided Wilcoxon rank sum tests. Lines = median; box = 25th–75th percentile; whiskers = 1.5 × interquartile range from box. **c** The rate of methylation gain is higher at H3K36me3 CGIs in HCT116 cells. Barplot of the median normalised rate of methylation gain for H3K36me3 marked CGIs and other CGIs methylated in HCT116 cells and DNMT KO cells ($n = 369$, 371 and 284 for K36me3 CGIs and 8285, 8284 and 6519 for other CGIs in HCT116, DNM3B KO and DNMT1 KO, respectively). The rate of methylation recovery was estimated by fitting linear models to the RRBS data for each CGI. *P*-values are from two-sided Wilcoxon rank sum tests. Source data are provided as a Source Data file.

H3K36me3 marked CGIs recovered DNA methylation significantly more rapidly than other CGIs at later time-points (Fig. 4b). Quantification of the individual normalised rates of methylation gain for each CGI confirmed that re-methylation was significantly faster at H3K36me3 marked CGIs than other CGIs (Fig. 4c). We then performed the same experiment in HCT116 cells lacking DNMT3B (DNMT3B KO cells) or DNMT1 (DNMT1 KO cells)[25]. Loss of DNMT3B attenuated the difference in re-methylation rate between H3K36me3 and other CGIs (Fig. 4c, Supplementary Fig. 4b) whereas the difference was exacerbated in DNMT1 KO cells (Fig. 4c, Supplementary Fig. 4c). This is consistent with the difference in re-methylation kinetics observed in HCT116 cells being caused by differential de novo DNMT activity and suggests that H3K36me3 marked CGIs are preferential targets of de novo DNMT activity in colorectal cancer cells.

**H3K36me3 marked CGIs are methylated in colorectal tumours and the normal colon.** Given that our experimental data suggested that de novo DNMT activity in colorectal cancer cells is primarily targeted to CGIs marked by H3K36me3, we wanted to understand whether H3K36me3 patterns were remodelled in colorectal cancer to cause aberrant hypermethylation at some CGIs.

We defined colorectal tumour H3K36me3 marked CGIs by re-analysing H3K36me3 ChIP-seq data from a colorectal tumour[33]. Using TCGA Infinium methylation array data from 342 colorectal tumours and 42 normal colon samples[34], we found that the vast majority of H3K36me3 CGI probes were highly methylated in clinical specimens (Fig. 5a–c). In contrast, CGI probes associated with H3K4me3, a mark that repels DNMT3 enzymes[35], had significantly lower methylation levels in clinical colorectal tumours (Fig. 5b, c). Similar results were observed in an independent dataset of 216 colorectal tumours and 32 normal colon samples (Supplementary Fig. 5a)[36] and the same CGIs were also highly methylated in adenomas (Supplementary Fig. 5b)[37]. DKO cell DNMT3B targets (from Fig. 2a) were also significantly enriched in colorectal tumour H3K36me3 marked CGIs ($p < 2.2 \times 10^{-16}$, Fisher's exact test) and depleted in colorectal tumour H3K4me3 marked CGIs ($p < 2.2 \times 10^{-16}$, Fisher's exact test).

Our analysis also demonstrated that H3K36me3 marked CGIs were highly methylated in the normal colon samples (Fig. 5a–c). An analysis of 38 matched tumour-normal pairs from TCGA

confirmed that these CGIs had similar methylation levels in the normal colon and colorectal tumours (mean Pearson correlation = 0.826). The correlation between matched samples for H3K4me3 peaks was significantly lower (mean 0.694, $p = 2.90 \times 10^{-9}$, Wilcoxon rank sum test). Using data from the Roadmap Epigenomics project, we then asked if colorectal tumour H3K36me3 marked CGIs were also associated with H3K36me3 in the normal colon. Our analysis showed strong normal colon H3K36me3 signal at these CGIs and 758/1061 overlapped a normal colon H3K36me3 peak (71.4%, $p < 2.2 \times 10^{-16}$, Fishers exact test, Fig. 5d, e). In contrast, colorectal tumour H3K4me3 marked CGIs had significantly lower levels of H3K36me3 in the normal colon ($p < 2.2 \times 10^{-16}$, Wilcoxon rank sum test) and were significantly depleted in normal colon H3K36me3 peaks (4.21% overlapped, $p < 2.2 \times 10^{-16}$, Fisher's exact test, Fig. 5e).

Taken together, these analyses suggest that the CGIs that are marked by H3K36me3 and subject to high de novo DNMT in colorectal cancer cells, are highly methylated in both clinical colorectal tumours and the corresponding normal tissue. They also suggest that H3K36me3 patterns are not extensively remodelled at CGIs in colorectal tumours.

**Discussion**
Here we present the first comprehensive assessment of de novo DNMT activity at CGIs in colorectal cancer. Our results suggest that this is primarily directed towards gene body CGIs marked by H3K36me3, a histone modification associated with ongoing active transcription, and lower levels are targeted to aberrantly methylated CGIs (Fig. 6). We find that these CGIs are not only highly methylated in tumours but also the normal colon suggesting that de novo DNMTs target similar CGIs in both normal somatic tissues and cancers.

Our observation that the highest levels of de novo methylation at CGIs in cancer cells are targeted towards those associated with H3K36me3 parallel observations made in diverse systems. Both DNMT3A and DNMT3B possess a PWWP domain that binds H3K36me3[38]. In mouse embryonic stem cells Dnmt3b is primarily localised to H3K36me3[17] and Dnmt3b knockout leads to preferential loss of DNA methylation from H3K36me3 marked regions[39]. H3K36me3 is deposited by SETD2 through its association with RNA polymerase II[40]. Transcription-induced

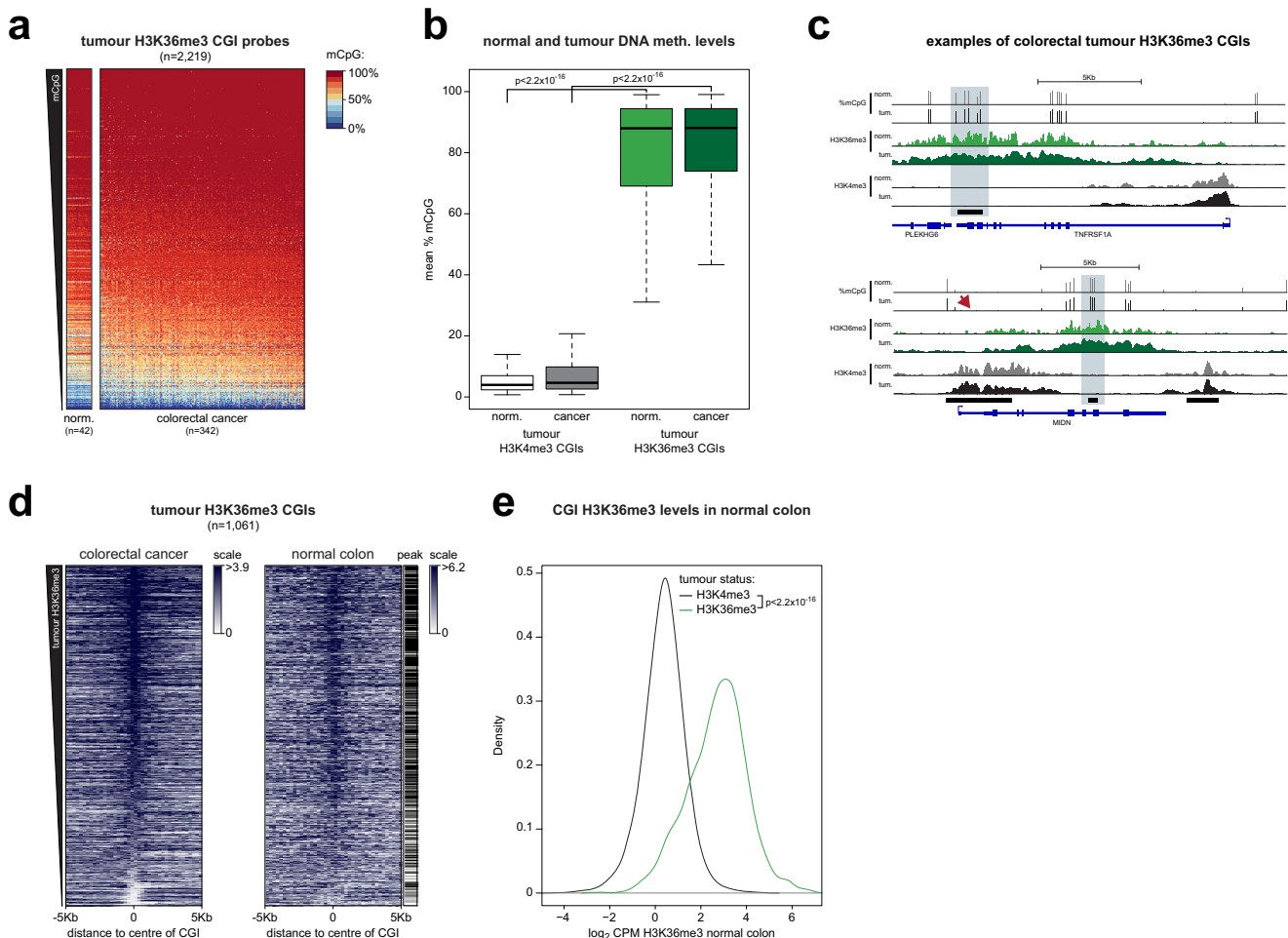

**Fig. 5 H3K36me3 marked CGIs are methylated in colorectal tumours and the normal colon. a** H3K36me3 marked CGIs are methylated in colorectal tumours. Heatmaps of methylation levels at probes located within H3K36me3 peaks and CGIs in colorectal tumours and adjacent normal colon samples. Both CpGs and samples are ordered by mean DNA methylation levels in colorectal tumours. norm. normal colon. **b** H3K36me3 marked CGIs are more highly methylated in colorectal tumours than H3K4me3 marked CGIs. Boxplot of the mean CpG methylation for probes located within CGIs and H3K36me3 or H3K4me3 peaks for 42 normal colon samples and 342 colorectal tumours. *P*-values are from two-sided Wilcoxon rank sum tests. Lines = median; box = 25th–75th percentile; whiskers = 1.5 × interquartile range from box. **c** Examples of colorectal tumour H3K36me3 marked CGIs. Genome browser plots showing mean DNA methylation levels along with H3K36me3 and H3K4me3 signal for both the normal colon and colorectal tumours. CGIs and genes are shown below the plots. CGIs marked by H3K36me3 in the colorectal tumour are indicated in light blue. Arrow indicates lack of methylation at an H3K4me3 marked CGI. norm. normal colon, tum. colorectal tumour. Scale for methylation data is 0 to 100%. **d** Colorectal tumour H3K36me3 CGIs are marked by H3K36me3 in the normal colon. Heatmaps of H3K36me3 ChIP-seq at colorectal tumour H3K36me3 CGIs. The left panel shows colorectal tumour H3K36me3 levels and the right panel normal colon H3K36me3 levels at the same CGIs. CGIs are ordered by mean H3K36me3 enrichment in the colorectal tumour. Those overlapping ChIP-seq peaks in the normal colon are indicated next to the heatmap (black = peak). **e** Colorectal tumour H3K36me3 CGIs have high levels of H3K36me3 in the normal colon. Density histogram of normalised normal colon H3K36me3 counts at CGIs marked by H3K36me3 and H3K4me3 in the colorectal tumour. *P*-value from two-sided Wilcoxon rank sum test. Source data are provided as a Source Data file.

deposition of H3K36me3 leads to Dnmt3b-dependent methylation of CGIs in mouse ES cells[41]. H3K36me3 is also associated with de novo methylation of imprinting control regions in mouse oocytes[42]. However, this is dependent on Dnmt3a and Dnmt3l but not Dnmt3b[43,44]. Previous work in DKO cells suggests DNMT3B is targeted to H3K36me3 in gene bodies but did not examine CGIs[45].

Overall, our results suggest that de novo DNMT activity is primarily targeted to H3K36me3 marked CGIs irrespective of the DNMT responsible but support a model in which the bulk of the de novo DNMT activity at H3K36me3 marked CGIs is dependent of DNMT3B in colorectal cancer cells. We also observe gains of DNA methylation at H3K36me3 marked loci when catalytically inactive DNMT3B is introduced into DKO cells. DNMT3A levels are increased in DKO cells and DNMT3A and B can interact[30]. A

previous study suggested that catalytically inactive DNMT3B may recruit DNMT3A to H3K36me3 marked gene bodies by comparing the kinetics of remethylation in cells with and without DNMT3A[45]. The structure of the catalytic domains of DNMT3B3 and DNMT3A2 bound to a nucleosome has also been solved[46]. These observations could also be explained by model where DNMT3A constitutively localises to H3K36me3 but that DNMT3A–DNMT3B hetero-complexes have higher catalytic activity[30]. This model is supported by a recent study showing that Dnmt3b3 can act as an accessory protein for Dnmt3a stimulating its catalytic activity at repetitive sequences[47]. Here, we directly confirm that DNMT3A is also more efficiently recruited to H3K36me3 marked CGIs in the presence of DNMT3B. A recent study demonstrates that Dnmt3a's PWWP has greater affinity for H3K36me2[48] potentially explaining why it is is less efficiently

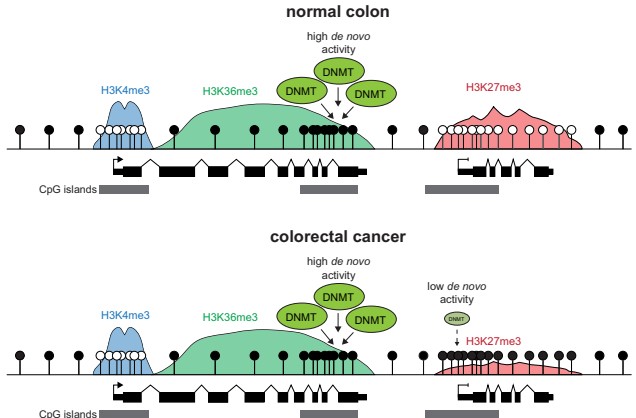

**Fig. 6 De novo DNA methyltransferase activity in colorectal cancer is directed towards H3K36me3 marked CpG islands.** Graphical model of the distribution of de novo DNMT activity at CGIs in colorectal cancer. Our data support a model whereby the CGIs subject to the highest levels of de novo DNMT activity in both the normal and cancerous colon are those marked by H3K36me3. Although CGIs marked by H3K27me3 in the normal colon become aberrantly methylated in colorectal cancer[4–6], our results suggest that this is associated with inefficient de novo methylation. H3K27me3 is lost from these CGIs when they become aberrantly methylated[83].

recruited to H3K36me3 marked CGIs than DNMT3B. Part of the de novo DNMT activity we measure at H3K36me3 marked CGIs in colorectal cancer cells could therefore be mediated by DNMT3A rather than DNMT3B but is primarily dependent on DNMT3B for recruitment.

Current models suggest that DNMTs are targeted to aberrantly methylated CGIs in colorectal cancer by transcription factors[15,16]. Aberrantly methylated CGIs are generally repressed in normal cells and associated with H3K27me3 and polycomb repressive complexes rather than H3K36me3[4–6]. Our results in this study suggest that the sequences of aberrantly methylated CGIs inefficiently recruit de novo DNMTs and are inconsistent with models of de novo methylation based on recruitment by transcription factors. A similar assay has previously been used to demonstrate sequence-specific programming of DNA methylation at CGIs by transcription factors in mouse ES cells[18]. Our observations suggest that aberrant CGI hypermethylation could occur through an inefficient, slow process associated with low de novo DNMT activity. In support of this hypothesis, the aberrantly methylated GSTP1 CGI promoter acquires little methylation when it is ectopically introduced into prostate cancer cells but gains of methylation were stimulated by prior in vitro seeding of low-level DNA methylation[49]. A recent study has shown that lowly expressed H3K27me3 marked CGIs gain methylation in normal mouse tissues when high ectopic levels of the embryonic active form of Dnmt3b, Dnmt3b1, are expressed[50]. However, the gains observed were not specific to the orthologues of those genes methylated in human tumours and the relative degree of Dnmt3b targeting of H3K36me3 marked loci was not assessed[50]. The relevance of DNMT3B overexpression in human cancer has also been questioned and apparent upregulation is suggested to reflect the greater proportion of cycling cells in tumour tissues[51,52]. The mechanism by which polycomb marked CGIs aberrantly gain methylation in cancer remains unclear[2]. It has been proposed that TET dysfunction mediated by mutations or hypoxia underpins this epigenetic switch[53,54]. Gains associated with TET-dysfunction could be expected to accumulate through the failure to remove aberrantly placed DNA methylation and thus could

occur despite a lack of strong targeting of these CGIs by de novo DNMTs.

Here we have used established cancer cell lines that do not model early stages of transformation. It is possible that a wave of de novo DNMT activity is targeted to aberrantly methylated CGIs specifically at the point of transformation paralleling the wave of genome-wide de novo methylation observed during early development[55]. If the signals causing this wave of de novo DNMT activity were subsequently lost in advanced tumours, they would not be ascertained in experiments using established cancer cell lines. However, oncogenic mutations are proposed to be the signal responsible for instructing de novo methylation[15,16] and these persist in advanced cancers and cell lines. Both HCT116 and RKO cells also have the ability to de novo methylate retro-viral DNA sequences[21] and we observe some de novo methylation at ectopic CGIs in our experiments. A further potential limitation of the use of cell lines is that DNA methylation patterns are altered by cell culture[56]. Little correspondence is reported between global methylation patterns in clinical ependymomas and cultured cell lines[57]. However, we have previously shown that aberrantly methylated CGIs identified in breast cancer cell lines are also identified in clinical tumours[5] and here we have focused on CGIs whose methylation is observed in vivo.

H3K36me3 has not been extensively examined in colorectal cancer but gains of DNA methylation in cancer have previously been associated with transcription across CGIs. In colorectal and breast tumours the TFPI2 promoter is aberrantly hypermethylated in association with transcription originating from a nearby LINE-1 promoter[58]. Also Lynch syndrome can be caused by the constitutive hypermethylation of the tumour suppressor MSH2 associated with read-through transcription from the upstream gene caused by a genetic deletion[59]. Given the association of SETD2 with RNA polymerase II[40], these cases of transcription-associated hypermethylation would be expected to be associated with H3K36me3 deposition as observed in experiments examining the effect of transcription across CGIs in differentiating mouse ES cells[41]. We find that colorectal tumour H3K36me3 marked CGIs are also associated with H3K36me3 and high levels of DNA methylation in the normal colon. This suggests that H3K36me3 patterns are not extensively remodelled in colorectal tumours compared to normal tissue and that transcription-coupled deposition of H3K36me3 is an infrequent cause of aberrant CGI hypermethylation in colorectal tumours. Instead, the recruitment of high de novo DNMT activity to H3K36me3-marked gene body CGIs might serve to prevent spurious transcription from these potential promoter sequences interfering with the expression of the associated genes[1,39].

Taken together our study suggests that the targeting of de novo DNMT activity to CGIs in colorectal cancer is surprisingly similar to that in normal cells and is predominantly centred on CGIs marked by H3K36me3.

## Methods

**Cell culture**. HCT116, DNMT1 KO, DNMT3B KO, and DKO cells were gifts from B. Vogelstein[24,25]. Cells were cultured in McCoy's 5A medium (Gibco). RKO cells were cultured in Dulbecco's Modified Eagles Medium (Sigma Aldrich). Both were supplemented with 10% foetal calf serum (Life technologies) and penicillin–streptomycin antibiotics at 140 and 400 μg/ml, respectively.

**Generation of plasmid constructs**. To create piggyBac transposon constructs containing CGIs, we amplified the CGIs from HCT116 cell genomic DNA and cloned them into the pEGFP-N2-pB-min containing piggyBac terminal repeats. To generate the plasmid pEGFP-N2-pB-min, we removed the multiple cloning site (MCS) of the pEGFP-N2 plasmid (Clontech) by BamHI-NheI double digestion, Klenow-mediated blunt ending and ligation. We then added a new MCS flanked by the minimal terminal repeats sufficient for PiggyBac transposition[60,61]. The new MCS was generated from two commercially synthesised oligonucleotides (IDT, oligo sequences in Supplementary Table 4) and inserted into the vector by In-

Fusion cloning (TakaraBio). CGI sequences were amplified using specific primers with 15 bp overhangs (Supplementary Table 4 for primer sequences) to facilitate In-Fusion cloning into pEGFP-N2-pB-min following the manufacturer's instructions.

To create the piggyBac DNMT3A and DNMT3B expression vectors, the DNMT3A1 and DNMT3B2 sequence from pcDNA3/Myc-DNMT3A1 and pcDNA3/Myc-DNMT3B2 plasmids (Addgene plasmids 36942 and 35521, gifts from A. Riggs)[32] was subcloned using XbaI and BamHI into the pCG plasmid (a gift from N. Gilbert) downstream of the T7 tag. The catalytically inactive point mutations C710S and C631S[62] were introduced using the QuikChange II site-directed mutagenesis kit (Agilent) to create T7-DNMT3A1 catalytic dead (CD) and T7-DNMT3B2-CD, respectively. The T7-tagged DNMTs were then cloned into pB530A-puroVal2 using BamHI and EcoRI (weak-expression vector with EF-1α promoter), and in PB-CGIP by swapping eGFP using AgeI and NotI (strong expression vector with CAG promoter, a gift from M. McCrew)[63]. pB530A-puroVal2 was previously created by substituting the copepod GFP from the pB530A plasmid (System Biosciences) with a Puromycin resistance gene.

**Ectopic integration of CGIs.** HCT116 cells were transfected using FuGENE HD transfection reagent (Promega). To ectopically integrate CGIs, mCherry expressing piggyBac transposase (gift from W. Akhtar)[64] was co-transfected with piggyBac CGI transposons into HCT116 cells in 1:1 ratio. After 48 h, cells expressing both GFP and mCherry were selected by FACS and expanded for 4 weeks to dilute out free plasmid. DNA was purified with phenol–chloroform extraction and ethanol precipitation. The methylation levels of CGIs at native and ectopic locations were then assessed by bisulfite PCR (primers listed in Supplementary Table 4). RKO cells were transfected similarly except Lipofectamine 2000 (Invitrogen) was used.

**DNMT3 expression in cells.** HCT116, DNMT3B KO and DKO cells were transfected using FuGENE HD (Promega). DNMT3A and DNMT3B expression constructs were co-transfected with piggyBac transposase. After 48 h, cells stably expressing DNMT3A/B were selected with 1.5 μg/ml puromycin (Thermo Scientific). Cells were expanded in the presence of puromycin for 20 days before genomic DNA was extracted using DNeasy Blood & Tissue Kit (QIAGEN) following the manufacturer's instructions or cells were used for ChIP. RKO cells were transfected similarly except Lipofectamine 2000 (Invitrogen) was used.

**Bisulfite PCR.** 500 ng genomic DNA was bisulfite converted with the EZ DNA Methylation-Gold kit (Zymo Research) according to the manufacturer's instructions. PCR was conducted using FastStart PCR Master Mix (Roche) or EpiTaq HS (TaKaRa), purified using the QIAquick PCR purification kit (QIAGEN) and sub-cloned into pGEMT-easy (Promega). Primer sequences are listed in Supplementary Table 4. Individual positive bacterial colonies were Sanger sequenced using SP6 or T7 sequencing primers (Supplementary Table 4) and analyzed using BIQ Analyzer software (for the HCT116 and RKO piggyBAC CGI experiments)[65] or BISMA (for all other experiments)[66]. To generate figures, results were then extracted from the BIQ or BISMA output HTML files using custom R scripts (available from: https://github.com/sproul-lab/masalmeh_et_al_2019_paper). Clones with ≥25% sequencing errors were excluded from analyses.

**5-aza-2′-deoxycytidine time-course.** Cells were plated at ~20% confluency. Next day cells were treated with freshly prepared 1 μM 5-aza-dC (Sigma Aldrich) for 24 h. Cell pellets were collected at different time points following treatment and genomic DNA was extracted using DNeasy Blood & Tissue Kit (QIAGEN) following the manufacturer's instructions.

**Western blotting.** Whole-cell extracts were obtained by sonication in UTB buffer (8 M urea, 50 mM Tris, pH 7.5, 150 mM β-mercaptoethanol) and quantified using $A_{280}$ from a NanoDrop (Thermo Scientific). 40–50 μg of protein was analysed by SDS–polyacrylamide gel electrophoresis (SDS–PAGE) using 3–8% NuPAGE Tris-Acetate or 4–12% NuPAGE Bis-Tris protein gels (Life Technologies) and transferred onto nitrocellulose membrane using Xcell Sure Lock Mini Cell electrophoresis tanks (Novex) in 2.5 mM tris-base, 19.2 mM glycine and 20% methanol. Immunoblotting was performed following blocking in 10% Western blocking reagent (Roche) using antibodies against DNMT3B (Cell Signalling Technology, D7O7O, 1:1000), GAPDH (Cell Signalling Technology, 14C10, 1:1000), DNMT3A (Cell Signalling Technology, 2160, 1:500) and the T7-tag (Cell Signalling Technology, D9E1X, 1:2000). Images were acquired with ImageQuant LAS 4000 following incubation with HRP conjugated goat anti-rabbit IgG (Invitrogen, A16110, 1:1000).

**Generation of CRISPR/Cas9-edited HCT116 cell line.** A guide RNA was designed using the CRISPR design web tool (http://crispr.mit.edu/) with the corresponding oligonucleotide cloned into pSpCas9(BB)-2A-GFP (pX458, Addgene Plasmid 48138, a gift of F. Zhang). The donor template for homology-directed repair was created by amplifying 3 single stranded DNA oligonucleotides (IDT Ultramers). The external sequences are 125 bp homology arms flanking each side of DNMT3B starting codon; the internal oligonucleotide is a triple T7 tag sequence.

The amplified donor template was then sub-cloned into pGEMT-easy (Promega). The vectors containing the gRNA sequence and the donor template were co-transfected using FuGENE HD transfection reagent (Promega). GFP positive cells were selected by FACS 48 h after transfection and plated at clonal density. Individual colonies were grown up, screened by PCR and the positive clone validated by Sanger sequencing. The positive clone carries one copy of DNMT3B with N-terminal triple-T7-tag. The second DNMT3B allele has been disrupted by the integration of a portion of the pSpCas9(BB)-2A-GFP plasmid downstream the 4th codon of DNMT3B. This is predicted to cause a frameshift with a severely truncated product on the second allele or be subject to nonsense-mediated decay. Primers and oligonucleotides used are listed in Supplementary Table 4.

**Chromatin immunoprecipitation.** For T7-DNMT3A1 and T7-DNMT3B2 ChIP experiments, $1 \times 10^7$ cells were harvested, washed and crosslinked with 1% methanol-free formaldehyde in PBS for 8 min at room temperature. Crosslinked cells were lysed for 10 min on ice in 50 μl of lysis buffer (50 mM Tris-HCl pH8, 150 mM NaCl, 1 mM EDTA, 1% SDS) freshly supplemented with proteinase inhibitor (Sigma-Aldrich). IP dilution buffer (20 mM Tris-HCl pH8, 150 mM NaCl, 1 mM EDTA, 0.1% Triton X-100) freshly supplemented with proteinase inhibitor, DTT and PMSF was added to the samples to reach a final volume of 500 μl. As prolonged sonication caused T7-DNMT3B degradation, chromatin was fragmented using Benzonase as recommended by Pchelintsev et al.[67]. Briefly, samples were sonicated on ice with Soniprep 150 twice for 30 s to break up nuclei. Then 200U of Benzonase Nuclease (Sigma) and $MgCl_2$ (final concentration 2.5 mM) were added and samples were incubated on ice for 15 min. The reaction was blocked by adding 10 μl of 0.5 M EDTA pH 8. Following centrifugation for 30 min at 18,407 g at 4 °C, supernatants were collected and supplemented with Triton X-100 (final concentration 1%) and 5% input aliquots were retained for later use. Protein A dynabeads (Invitrogen) previously coupled with 10 μl of T7-Tag antibody per $1 \times 10^6$ cells (D9E1X, Cell Signalling) in blocking solution (1xPBS, 0.5% BSA) were added and the samples incubated overnight under rotation at 4 °C. Beads were then washed for 10 min at 4 °C with the following buffers: IP dilution buffer 1% Triton X-100 (20 mM Tris-HCl pH 8, 150 mM NaCl, 2 mM EDTA, 1% Triton X-100), buffer A (50 mM Hepes pH 7.9, 500 mM NaCl, 1 mM EDTA, 1% Triton X-100, 0.1% Na-deoxycholate, 0.1% SDS), buffer B (20 mM Tris pH 8, 1 mM EDTA, 250 mM LiCl, 0.5% NP-40, 0.5% Na-deoxycholate), TE buffer (1 mM EDTA pH 8, 10 mM Tris pH 8). Chromatin was eluted by incubating the beads in extraction buffer (0.1 M $NaHCO_3$, 1% SDS) for 15 min at 37 °C. To reverse the cross-linking Tris-HCl pH 6.8 and NaCl were added to final concentrations of 130 mM and 300 mM respectively. IP samples were then incubated at 65 °C for 2 h for ChIP-qPCR experiments and overnight for ChIP-Rx-seq experiments. Samples were then incubated at 37 °C for 1 h after addition of 2 μl of RNase Cocktail Enzyme Mix (Ambion). Then 40 μg of Proteinase K (Roche) were added, followed by 2 h incubation at 55 °C. Input material was similarly de-crosslinked. Samples were purified with the MinElute PCR purification kit (QIAGEN).

qPCR was performed using Lightcycler 480 Sybr Green Master (Roche) on a Light Cycler 480 II instrument (Roche). Primers used are listed in Supplementary Table 4. Enrichment was calculated compared to input DNA. Delta $C_t$ was calculated using input $C_t$ values adjusted to 100% and assuming primer efficiency of 2.

For DNMT3B ChIP-Rx-seq, 20 μg of Spike-in chromatin (ActiveMotif 53083) was added to each sample prior to sonication. 2 μl of spike-in antibody per sample (ActiveMotif 61686) was also added in a ratio 1:5 versus the T7-antibody. A similar protocol was used for H3K36me3 ChIP ChIP-Rx-seq experiments except: $0.5 \times 10^7$ cells were harvested and crosslinked with 1% methanol-free formaldehyde in PBS for 5 min at room temperature. Crosslinked Drosophila S2 cells were spiked into samples before sonication at a ratio of 20:1 human to Drosophila cells. Following nuclei rupture by sonication on ice with Soniprep 150, chromatin was fragmented using Bioruptor Plus sonicator (Diagenode) for 40 cycle (30 s on/30 s off on high setting at 4 °C). 2 μl of H3K36me3 antibody per $1 \times 10^6$ cells (ab9050, Abcam) was used for immunoprecipitation.

For ChIP-Rx-seq, libraries were prepared using the NEBNext® Ultra™ II DNA Library Prep Kit for Illumina (E7645) according to the manufacturer instructions. Barcoded adapters (NEBNext® Multiplex Oligos for Illumina® Index Primers Set 1, E7335) were used. For H3K36me3 ChIP-Rx-seq, adapter-ligated DNA was size selected for an insert size of 150 bp using AMPure XP beads. ChIP-Rx-seq libraries were sequenced using the NextSeq 500/550 high-output version 2.5 kit (75 bp paired end reads, DNMT3B or 75 bp single end reads, H3K36me3). Libraries were combined into equimolar pools to run within individual flow cells. Sequencing was performed by the Edinburgh Clinical Research Facility.

**Global measurement of DNA methylation by mass-spectrometry.** 1 μg genomic DNA was denatured at 95 °C for 10 min in 17.5 μL water. DNA was then digested to nucleotides overnight at 37 °C with T7 DNA polymerase (Thermo Scientific). The reaction was inactivated by incubating at 75 °C for 10 min. Samples were then centrifuged for 45 min at >12,000 g and the supernatant transferred into new tubes for analysis. Enzyme was removed by solvent precipitation. The samples were adjusted back to initial aqueous condition and volume and LC-MS was performed on a Dionex Ultimate 3000 BioRS / Thermo Q Exactive system, using a Hypercarb 3 μm × 1 mm × 30 mm Column (Thermo 35003-031030) and gradient from

20 mM ammonium carbonate to 2 mM ammonium carbonate 90% acetonitrile in 5 min. Data were acquired in negative mode, scanning at 70 k resolution from 300 to 350 m/z. Extracted ion chromatograms were analysed using Xcalibur (Thermo Scientific, v2.5-204201/2.5.0.2042) to extract peak intensities at the m/z values expected for each nucleotide (based on annotation from METLIN[68] following manual inspection to ensure that they were resolved as clear single peaks. The % of 5-methylcytosine present in the sample was calculated as the ratio of the area under the 5-methylcytosine peak to the area under the guanine peak.

**Statistical analysis**. Statistical testing was performed using R v3.4.2. All tests were two-sided, unless otherwise stated. Further details of specific analyses are provided in the relevant methods sections below.

**CGI definition**. CGI annotation was taken from Illingworth et al.[69]. Overlapping CGI intervals were merged using BEDtools (version 2.27.1)[70] before they were converted to hg38 positions using the UCSC browser liftover tool. Non-autosomal CGIs were excluded from the analysis as were CGIs located in ENCODE blacklist regions[27]. CGIs were annotated relative to genes using BEDTools to overlap them with ENSEMBL protein coding genes (Ensembl Release 98/GCRh38). CGIs were annotated as being located at a transcription start site (TSS) if they overlapped a protein coding TSS and as located in a gene body if they overlapped a transcript but not a TSS. The remaining CGIs which did not overlap a TSS or transcript were annotated as intergenic.

To analyse GO-terms associated with DNMT3B target CGIs while reducing biases associated with the uneven distribution of genes in the genome[71], CGIs were mapped to protein coding genes using BEDtools. CGIs overlapping TSSs were assigned to the genes associated with those TSSs. If a CGI did not overlap a TSS but lay within a transcript, it was assigned to those gene(s) and if not to the closest TSS (s). CGIs were then mapped to Biological Process, Molecular Function and Cellular Compartment GO-terms through their assigned genes using ENSEMBL Biomart. All parental terms were identified using R scripts and the Bioconductor GO.db package (version 3.4.1). For each GO term, statistical enrichment for term-associated CGIs in the DKO target list versus a background list of all CGIs was tested by Benjami–Hochberg corrected Fisher's exact test, with an FDR rate of 0.05. Terms with <10 CGIs in the background list were excluded from analysis. GO terms were filtered for semantic similarity (Bioconductor packages GOSemSim version 2.14.2 and org.Hs.eg.db version 3.11.4)[72]. Similar terms were defined on the basis of Wang similarity >0.7[72], using a modified simplify function from the clusterProfiler (version 3.16.1) Bioconductor package[73]. For each group, the term with the lowest adjusted p-value was retained.

**RRBS data generation**. Genomic DNA was extracted from cells using DNeasy Blood & Tissue Kit according to the manufacturer's protocol with some modifications. RNase A/T1 Cocktail (Ambion AM2286) was added to proteinase K and samples were incubated at 37 °C for 1 h to remove RNA. At the final step, DNA was eluted with ddH$_2$O instead of AE buffer. DNA was quantified by Nanodrop and Qubit.

200 ng of purified DNA samples were processed using the Ovation RRBS Methyl-Seq system kit (NuGen Technologies) according to instructions with modifications. Briefly, 0.5 ng unmethylated phage λ DNA (NEB) was spiked into each sample to allow assessment of bisulfite conversion efficiency. The methylation-insensitive restriction enzyme MspI was then used to digest the genomic DNA, and digested fragments were ligated to adapters. Adapter-ligated fragments were then repaired before bisulfite conversion with the Qiagen Epitect Fast Bisulfite Conversion kit. Bisulfite-treated adapter-ligated fragments were amplified by 9 cycles of PCR and purified using Agencourt RNAClean XP beads. Libraries were quantified using the Qubit dsDNA HS assay and assessed for size and quality using the Agilent Bioanalyzer DNA HS kit.

Sequencing was performed using the NextSeq 500/550 high-output version 2 kit (75 bp paired end reads) on the Illumina NextSeq 550 platform. As instructed for the NuGen RRBS kit, 12 bp index reads were generated to sequence the Unique Molecular Identifiers (UMI) in addition to the index present in the adaptors. Libraries were combined into equimolar pools and run on a single flow cell. 10% PhiX control library (Illumina v3 control library) was spiked in to facilitate sequencing by generating additional sequence diversity. Library preparation and sequencing was performed by the Edinburgh Clinical Research Facility.

**RRBS data processing**. Raw Illumina sequencing output from the NextSeq (bcl files) were converted to paired FASTQ files without demultiplexing using bcl2fastq and default settings (v2.17.1.14). These FASTQ files were then demultiplexed using custom python scripts considering indexes with perfect matches to the sample indexes. The different lanes for each sample were then combined.

Sequencing quality was assessed with FASTQC (v0.11.4). Low quality reads and remaining adaptors were removed using TrimGalore (v0.4.1, Settings: --adapter AGATCGGAAGAGC --adapter2 AAATCAAAAAAAC). NuGen adaptors contain extra diversity bases to facilitate sequencing. These were removed using the trimRRBSdiversityAdaptCustomers.py Python script provided by NuGen (v1.11). The paired end reads were then aligned to the hg38 genome using Bismark (v0.16.3 with Bowtie2 v2.2.6 and settings: -N 0 -L 20)[74,75] before PCR duplicates were

identified and removed using the 6 bp UMIs present in the index reads and the nudup.py Python script supplied by NuGen (v2.3). Aligned BAM files were processed to report coverage and number of methylated reads for each CpG observed. Forward and reverse strands were combined using Bismark's methylation extractor and bismark2bedgraph modules with custom Python and AWK scripts (available from: https://github.com/sproul-lab/masalmeh_et_al_2019_paper).

Processed RRBS files were assessed for conversion efficiency based on the proportion of methylated reads mapping to the λ genome spike-in (>99.5% in all cases). For summary of RRBS alignment statistics see Supplementary Table 5. BigWigs were generated from RRBS data using CpGs with coverage ≥5.

**Analysis of RRBS data**. CGI methylation levels were calculated as the weighted mean methylation using the observed coverage of CpGs within the CGI (unconverted coverage/total coverage). Only CpGs with a total coverage ≥10 in all samples were considered in each analysis. CGIs significantly gaining methylation upon expression of DNMT3B in DKO cells were defined as those with a Benjamini Hochberg adjusted Fisher's exact test p-value <0.05 and where ≥20% methylation was gained.

**Analysis of re-methylation kinetics following 5-aza-2′-deoxycytidine treatment**. If re-methylation following 5-aza-dC treatment was entirely driven by outgrowth of unaffected cells, we expect the methylation trajectory of each CGI to be proportional to its initial methylation level (Supplementary Fig. 6). In order to test this possibility, we normalised HCT116 methylation data for all CGIs with day 0 mean CGI methylation >50% over the methylation value at day 0. Two models were fitted to this normalised data using time-points between day 3 and day 22 inclusive. The null hypothesis that re-methylation is due to outgrowth was represented by the mixed linear model shown in Eq. 1. The hypothesis that each CGI is remethylated at a different rate was represented by a mixed linear model accounting for a different trajectory for each CGI shown in Eq. 2.

$$m \sim t + m_{CGI} \tag{1}$$

$$m \sim t + m_{CGI} + s_{CGI} \times t \tag{2}$$

where $m$ is mean normalised methylation, $t$ is the time, $m_{CGI}$ is the random intercept for each CGI, and $s_{CGI}$ is the random slope for each CGI. Comparing the models with ANOVA reveals that the probability that Eq. 1 is a better fit to the data is $p = 1.31 \times 10^{-25}$, strongly suggesting that the CGIs do not recover methylation at the same rate. The R function lmer from the lme4 package (v1.1) was used for this analysis.

In order to investigate the trajectories of each CGI independently from one another, a linear model was fitted to the normalised methylation of each individual CGI at time points between day 3 and day 22 inclusive. The slopes of these individual linear models are equal to the fraction of methylation recovered per day. This analysis was repeated for the DNMT3B KO and DNMT1 KO cell lines.

**Processing of ChIP-seq data**. All ChIP-seq experiments were processed as follows. Read quality was checked using FASTQC (v0.11.4, https://www.bioinformatics.babraham.ac.uk/projects/fastqc), with low quality reads and adaptors removed using trim-galore with default settings (version 0.4.1). Reads were aligned to hg38 using bowtie 2 (v2.3.1, with settings: -N 1 -L 20 --no-unal)[74]. Multi-mapping reads excluded using SAMtools (v1.6, with settings: -bq 10)[76] and PCR duplicates excluded using SAMBAMBA (v0.5.9)[77]. For paired end data, additional settings were used during alignment to remove discordant reads: --no-mixed --no-discordant -X 1000. ChIP-Rx-seq reads were aligned to a combination of hg38 and dm6 genomes. For summary of ChIP-seq alignment statistics see Supplementary Table 6.

Processed ChIP-seq data from the normal colon and HCT116 cells were downloaded from ENCODE[27] as aligned BAM files and replicated peak BED files. The corresponding annotated input control was also downloaded for each sample. bigWig files for visualisation were downloaded from ENCODE. Data for colorectal tumour H3K36me3 and H3K4me3 ChIP-seq[33] was downloaded as FASTQ files from ENA and aligned as above.

Tracks for data visualisation were generated using Deeptools (v3.2.0)[78]. Counts per million normalised tracks were generated using the bamCoverage function (settings: --normalizeUsing CPM) with the default bin size of 50 bp. The mean of replicate tracks was calculated using the bigwigCompare function (settings: --operation mean). For the single-ended H3K36me3 ChIP-seq, the estimated fragment length of 150 bp was used. For the paired-end T7-DNMT3B ChIP-seq, the actual fragment size was used. For the colorectal tumour ChIP-seq, tracks were instead generated using Homer (v4.8)[79]. Aligned BAMs were converted to tag directories setting the fragment length to 180 bp and converted to bigWig files using Homer's makeUCSCfile function after filtering with the removeOutOfBounds.pl function. ChIP-seq peaks were called from tag directories using Homer's findPeaks function (settings: -style histone) and an input chromatin control sample.

**Analysis of ChIP-seq data**. Normalised read counts for CGIs were derived from ChIP-seq by first counting the number of reads or fragments overlapping each CGI using BEDtools' coverage function. For heatmaps of ChIP-seq data, non-

overlapping windows of 250 bp were defined centred on the midpoint of each CGI and read or fragment counts/window similarly calculated. In both cases, read counts were scaled to counts per 10 million based on total number of mapped reads/sample and divided by the input read count to provide a normalised read count. To prevent windows with zero reads in the input sample generating a normalised count of infinity, an offset of 0.5 was added to all windows prior to scaling and input normalisation. Regions where coverage was 0 in all samples were removed from the analysis.

ChIP-Rx was analyzed similarly before samples were scaled using a normalisation factor generated from the number of reads mapping to the spike-in D. melanogaster genome. Reads mapping to the *D.melanogaster* genome in each ChIP and input sample were first scaled to reads per $1 \times 10^7$. The scaling factor was then calculated as the ratio of the scaled D. melanogaster reads in two ChIP samples over their respective ratio from the input samples; that is scaling factor, S, for sample $n$ compared to reference sample ref: $S_n = (dRPTM_{ChIP-n} / dRPTM_{ChIP-ref}) / (dRPTM_{IN-n} / dRPTM_{IN-ref})$, where $dRPTM = D. melanogaster$ reads per $1 \times 10^7$ for ChIP and input (IN) runs, respectively (modified from published method to take account of the presence of an input sample)[28]. This was separately applied to each biological replicate of the experiments.

Where multiple replicates were available, the mean was calculated for each CGI or window. Colour scales for ChIP-seq heatmaps range from the minimum to the 90% quantile of the normalised read count. ChIP-seq peaks were overlapped with CGIs using BEDtools intersect and tested for statistical enrichment with Fisher's exact tests versus all CGIs included in each analysis. To statistically test differences in histone modification levels, normalised read depths across CGIs were compared using a Wilcoxon rank sum test.

**Infinium array data processing**. All available colorectal tumour and normal colon data were downloaded from TCGA Genomic Data Commons as raw Illumina 450k IDAT files on 13/9/18[34]. These were processed using the ssNoob approach from the Bioconductor package *minfi* (v1.22.1) to derive beta values and detection *p*-values (beta threshold = 0.001)[80,81]. Individual beta values were excluded where detection *p*-value was >0.01. Non-CG probes were also excluded from the analysis. Samples that did not represent primary tumour or adjacent normal colon tissue were excluded using TCGA sample type codes as were samples where ≥1% of probes failed the detection *p*-value threshold. Where replicates existed for these tumour and normal samples, the mean was calculated for each probe. This left 342 tumour samples and 42 adjacent normal colon samples and included 38 matched tumour-normal pairs. Infinium probe locations in the hg38 genome build were taken from Zhou et al.[82]. Probes categorised as overlapping common SNPs or having ambiguous genome mapping in that paper were excluded from the analysis (Zhou et al. general masking annotation). A second dataset of 216 colorectal tumours and 32 adjacent normal tissues[36] was downloaded as IDAT files from Array Express (accession: E-MTAB-7036) and processed similarly. Raw IDAT files were not available for the ademona dataset[37] so the author's processed data were downloaded from the NCBI's Gene Expression Omnibus (GEO, accession: GSE48684) as a series matrix. These values were then treated similarly to the other datasets.

To define the frequency of aberrant methylation at selected CGIs, the mean methylation of each CGI in each sample was calculated from the probes contained within it. A CGI was then defined as methylated in a given sample if its beta >= 0.3 using a previously published threshold[4].

**Reporting summary**. Further information on research design is available in the Nature Research Reporting Summary linked to this article.

## Data availability

All sequencing data generated in this study are available in GEO under accession GSE158406. Publicly available tumour methylation data from tumours are available from GEO under accession GSE48684 and Array Express under accession E-MTAB-7036. All other relevant data supporting the key findings of this study are available within the article and its Supplementary Information files or from the corresponding author upon reasonable request. Source data are provided with this paper. A reporting summary for this Article is available as a Supplementary Information file. Source data are provided with this paper.

## Code availability

Custom scripts used in the analysis of data are available from: https://github.com/sproul-lab/masalmeh_et_al_2019_paper.

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

## Acknowledgements

We thank W. Bickmore, R. Meehan, P. Heyn, R. Illingworth, C. Tufarelli and G. Ficz for helpful discussions about the study and the Edinburgh Clinical Research Facility Genetics Core and the IGMM mass-spectrometry facility for technical support. This work has made use of the resources provided by the University of Edinburgh digital research services and the MRC IGMM compute cluster. D.S. is a Cancer Research UK Career Development fellow (reference C47648/A20837), and work in his laboratory is also supported by an MRC university grant to the MRC Human Genetics Unit. R.H.A.M. was funded by a HESPAL PhD scholarship from the British Council. I.K. is funded by a studentship from CRUK. K.P.P. and H.M.F. were funded by ERASMUS+ scholarships.

## Author contributions

R.H.A.M., F.T., C.R.R., K.M., H.D.S., I.K., K.P.P. and H.F. performed the experiments included in the manuscript. J.W. conducted mass spectrometry supported by AJF. R.C. conducted high-throughput sequencing supported by L.M. R.H.A.M., J.H. and D.S. conducted the computational analysis presented in the manuscript. D.S. planned and supervised the study and wrote the manuscript with input from R.H.A.M. and review by all authors.

## Competing interests

The authors declare no competing interests.
