## [Peer Review File · Nature Communications]

Reviewers' comments:

Reviewer #1 (Remarks to the Author):

Aberrant DNA hypermethylation of CpG islands (CGIs) is a common feature in cancer cells, which is implicated in silencing tumor suppressors. It is unknown why some CGIs, but not others, are hypermethylated. In this study, Sproul et al. provide evidence that abnormal de novo methylation is not sequence specific. Rather, hypermethylated CGIs correlate with high levels of H3K36me3 in cancer cells and normal cells.

While the observation is important, the study is preliminary overall.

Main issues:

1. The observation raises a critical question: Why do some CGIs abnormally gain H3K36me3 in cancer cells? While CGIs in gene bodies may gain H3K36me3 during transcriptional elongation, how about CGIs in promoter regions and intergenic regions?
2. The authors showed that re-expression of catalytically inactive DNMT3B isoforms in DNMT1/3B DKO HCT116 cells also results in gain of DNA methylation, suggesting that DNMT3A activity is important. A recent Nature paper reported that DNMT3A preferentially recognizes H3K36me2 and shapes DNA methylation in intergenic regions, whereas DNMT3B preferentially recognizes H3K36me3 and methylates CpGs in transcribed (genic) regions. The authors ought to compare the gain of methylation at CpGs and CGIs between cells re-expressing DNMT3B2 and those expressing inactive DNMT3B, taking into consideration of H3K36me2 as well.

Both points are related. I believe that additional experimental data or computational data that provide further insights into these questions will make the study a lot more significant.

Minor point:

3. The sentence "Transcription deposition of H3K36me3 leads to DNMT3B-dependent methylation of CGIs in mouse ES cells and is associated with de novo methylation of imprinting control regions in mouse oocytes" is confusing and inaccurate, because genetic studies have shown that DNMT3A and its co-factor DNMT3L, but not DNMT3B, are essential for methylation of ICRs in germ cells.

Reviewer #2 (Remarks to the Author):

Key results:

In order to maintain fidelity of DNA methylation during mitosis DNMT and TET families of enzymes work synergistically. Maintenance DNA methylation is undertaken by DNMT1 and de novo DNA methylation by DNMT3a/3b.

It is widely accepted that aberrant gain of DNA methylation at CpG islands occurs in colorectal cancer tumorigenesis, leading to altered gene expression of tumour suppressor and oncogenes. There is some evidence that CpG islands are targeted in a sequence dependant manner through de novo DNA methyltransferase activity.

This paper sets out to determine where de novo DNA methyltransferases target CpG islands in colorectal tumorigenesis using ectopically manipulated in vitro cell culture of one colorectal cell line (HCT116) and normal epithelia. Conceptually this is a very relevant area to investigate in tumorigenesis as epigenetic targeting and manipulation are becoming therapeutic options. The authors suggest that their findings demonstrate a low level of de novo methylation at CpG islands, and that when it does occur it is mainly at CpG islands which are marked by the histone modification H3K36me3 which is associated with transcriptional elongation. Furthermore it is suggested that as H3K36me3 marked CpG islands are highly methylated in normal colorectal epithelia and the HCT116 cell line, de novo DNA methylation in colorectal cancer tumorigenesis occurs at the same targets as normal colorectal tissue and is not altered in tumorigenesis.

Validity:

There are significant limitations in the approach that was used to test the experimental hypothesis. These limitations relate to the use of cell lines and the ability to translate these findings to in vivo systems and tissues.

1: Only one colorectal cell line was used for this work – when there are many available. The use of biological and technical replicates is not clear in the methods. There is one reference to biological replicates in the figure legend of figure 2e and a reference to technical replicates is made in the legend of Supplementary Figure 1. The authors need to clarify and include information regarding biological replicates.

2: Cell culture affects genome-wide DNA methylation – this has significant implications in the interpretation of the results presented in this paper and requires acknowledgement in the introduction and discussion. Rogers et al have described a lack of concordance between DNA methylation profiles of ependymoma cell lines and tissue (Oncotarget. 2018 Nov 23; 9(92): 36530–36541). This is particularly relevant when the authors conclude that their findings can be extrapolated to colorectal cancer tissue.

Originality and significance:

This is the first group to suggest these conclusions. However I have reservations regarding the ability to draw these conclusions from the work described alongside the limitations of using one colorectal cancer cell line and the effects of cell culture on genome-wide DNA methylation profiles. I do, however appreciate the lack of alternative models available to test the experimental hypothesis. Zhang et al used an inducible transgenic mouse model focusing on the impact of genome-wide de novo DNMT3b activity across a number of cell types – with different findings to those presented in this paper (eLife. 2018; 7: e40757. Published online 2018 Nov 23. doi: 10.7554/eLife.40757 PMID: PMC6251628). This work is not referred to in the paper and is relevant.

Data & methodology:

Within the limitations described previously, the cell line work, piggyBac and experimental approach are appropriate and described with sufficient detail to allow another group to undertake these experiments.

The description of the data analysis undertaken requires more detail – for example “Infinium array data processing” (pg 21) requires information about whether this is data from the Illumina HumMeth 27 or 450K arrays, what normalisation steps were taken to address batch effects etc.

Appropriate use of statistics:

Statistical tests and application are appropriate. However, description of error bars is missing in the legends of Figure 2d, 3b, 4b

Conclusions: Do you find that the conclusions and data interpretation are robust, valid and reliable?

Unfortunately I do not find the conclusions to be robust due to the limitations described previously regarding the use of one cancer cell line and the effects of genome-wide DNA methylation in cell culture.

Suggested improvements:

In order to have more certainty of the findings described in the paper I would suggest

- 1: Repeating the experiment with an alternative colorectal cancer cell line and normal colorectal tissue cell line.
- 2: Reference to and discussion around the effect of cell culture on genome-wide DNA methylation and the impact this has on interpretation of these results.
- 3: Clear indication of the use of biological and technical replicates and annotation of figure legends appropriately regarding error bars.
- 4: Including additional detail to the data analysis methodology for the Illumina HumMeth Infinium array data.

References:

The work of Zhang et al is not referenced and is relevant (eLife. 2018; 7: e40757. Published online 2018 Nov 23. doi: 10.7554/eLife.40757 PMID: PMC6251628).

Clarity and context:

The abstract and introduction are clear.

The conclusion is written clearly, but for reasons described previously I do not feel that the findings can be extrapolated to colorectal cancer tissue.

Reviewer #3 (Remarks to the Author):

This study uses the mismatch repair-deficient cell line HCT116 and its derivatives (DKO, DNMT3B-KO and DNMT1-KO) to investigate the mechanisms of CpG island (CGI) targeting by de novo DNMT activity. The findings disagree with prevailing models of sequence-specific (transcription factor-mediated) targeting, and propose a new, H3K36me3-based model, on the following evidence. First, results from a transposon-based approach for random, ectopic integration of selected CGIs into the HCT116 genome suggested that targeting of de novo DNMT activity is not sequence-specific. Second, introduction of DNMT3B into DKO cells revealed over 2,000 CGIs that gained methylation and the respective regions were marked by H3K36me3. Third, 5-aza-dC experiments showed that post-treatment remethylation was significantly faster at H3K36me3-marked CGIs compared to other CGIs. Fourth, re-analysis of TCGA-derived H3K36me3 ChIP-seq data from colorectal tumors and normal colon samples (including a set of matched normal-tumor pairs) showed that methylation levels were equally high in normal and tumor tissues, arguing against tumor-specific remodeling of H3K36me3 patterns at CGIs.

This work provides a comprehensive investigation into the mechanisms of de novo DNMT activity at CGIs in colorectal cancer. The observation that de novo DNMT activity is predominantly targeted at gene body CGIs marked by H3K36me3 is an important new finding supported by previous clues (references 17 and 36).

Comments:

1. As the authors state (Discussion, p. 10), the HCT116 colorectal cancer cell line models late stages of tumorigenesis after malignant transformation. Regarding the TCGA clinical colorectal tumor samples investigated, did they all represent carcinomas? Perhaps benign tumors (adenomas) were included as well?
2. Did the authors perform a closer analysis of the 2,238 CGIs that gained significant methylation in DKO cells upon DNMT3B introduction? How were the respective genes distributed according to different functional categories, for example?
3. Discussion on p. 10 ("Our observations suggest that aberrant CGI hypermethylation could occur through an inefficient, slow process associated with low de novo DNMT activity") and the right half

of Fig. 5: If de novo DNMT activity is low, as observed by the authors, what is the (likely) mechanism behind aberrant tumor-specific methylation of CGIs based on available evidence?

Response to Reviewers NCOMMS-19-29594 Masalmeh *et al.*

We thank the reviewers for recognizing the importance of our work and for their supportive and insightful comments that have helped us substantially improve the manuscript. Our institute was closed between the 20th March and 6th July due to the COVID-19 pandemic. We have now partially re-opened in line with social distancing rules in the UK and have completed experiments and analyses to address the points made by all reviewers. In particular, we demonstrate the reproducibility of our results in a further cell line, their relevance to clinical colorectal tumours and clarify the role of DNMT3A.

We enclose a revised copy of the manuscript with changes highlighted in blue.

In addition to your comments we received editorial advice in the decision letter (comments are highlighted in *blue italics* throughout this text):

Your manuscript entitled "De novo DNA methyltransferase activity in colorectal cancer is directed towards H3K36me3 marked CpG islands" has now been seen by 3 referees, whose comments are appended below. You will see from their comments copied below that while they find your work of considerable potential interest, they have raised quite substantial concerns that must be addressed. In light of these comments, we cannot accept the manuscript for publication, but would be interested in considering a revised version that addresses these comments in full.

Reviewer #1:

Aberrant DNA hypermethylation of CpG islands (CGIs) is a common feature in cancer cells, which is implicated in silencing tumor suppressors. It is unknown why some CGIs, but not others, are hypermethylated. In this study, Sproul et al. provide evidence that abnormal de novo methylation is not sequence specific. Rather, hypermethylated CGIs correlate with high levels of H3K36me3 in cancer cells and normal cells.

While the observation is important, the study is preliminary overall.

Main issues:

1. The observation raises a critical question: Why do some CGIs abnormally gain H3K36me3 in cancer cells? While CGIs in gene bodies may gain H3K36me3 during transcriptional elongation, how about CGIs in promoter regions and intergenic regions?

As the reviewer suggests, the CGIs targeted by *DNMT3B* in our experiments are primarily found in gene bodies as well as being marked by H3K36me3 (64.08% of targets, new analysis in Supplementary Fig. 2d).

The analyses we present in Fig. 5d,e suggest that little remodelling of H3K36me3 patterns occurs in colorectal cancer. However, a few examples of promoter CGI hypermethylation associated with abnormal transcription across them exist in the literature with relevance to colorectal cancer. At present it remains unclear whether these events are associated with H3K36me3. Given that our analyses and the literature suggest that gains of H3K36me3 at CGIs in colorectal tumours are rare, their full exploration would require a large study of clinical tumours something that is beyond the scope of the time constraints of a paper revision. We have, however, included discussion of the literature in the manuscript:

H3K36me3 has not been extensively examined in colorectal cancer but gains of DNA methylation in cancer have previously been associated with transcription across CGIs. In colorectal and breast tumours the TFPI2 promoter is aberrantly hypermethylated in association with transcription originating from a nearby LINE-1 promoter⁵⁶. Also Lynch syndrome can be caused by the constitutive hypermethylation of the tumour suppressor MSH2 associated with read-through transcription from the upstream gene caused by a genetic deletion⁵⁷.

2. The authors showed that re-expression of catalytically inactive DNMT3B isoforms in DNMT1/3B DKO HCT116 cells also results in gain of DNA methylation, suggesting that DNMT3A activity is important. A recent Nature paper reported that DNMT3A preferentially recognizes H3K36me2 and shapes DNA methylation in intergenic regions, whereas DNMT3B preferentially recognizes H3K36me3 and methylates CpGs in transcribed (genic) regions. The authors ought to compare the gain of methylation at CpGs and CGIs between cells re-expressing DNMT3B2 and those expressing inactive DNMT3B, taking into consideration of H3K36me2 as well.

In order to clarify the role of DNMT3A, we performed ChIP-seq for H3K36me2 in DKO cells as the reviewer suggests. However, despite using the same antibody as the Weinberg study referred to by the reviewer, the signal to noise ratio of our data was too low to include in the revised manuscript.

We therefore performed further experiments to examine the potential role of DNMT3A (included as Supplementary Fig. 2i-k). These show that overexpression of DNMT3A in DKO cells results in gains of methylation at DNMT3B target loci. However, these gains were less than observed when expressing DNMT3B. This suggests that DNMT3A can methylate these loci but does so inefficiently. We then confirmed that DNMT3B can facilitate the recruitment of DNMT3A to our DNMT3B target loci using ChIP. Higher levels of DNMT3A were observed at these loci in HCT116 cells than the DNMT3B KO derivatives. This demonstration of DNMT3B-mediated recruitment of DNMT3A strongly supports the role of DNMT3A in the gains of DNA methylation we observe at H3K36me3 marked CGIs when we express catalytically-dead DNMT3B. We have also added discussion of the implications of the Weinberg study for our results to the manuscript:

A recent study demonstrates that Dnmt3a's PWWP has greater affinity for H3K36me2⁴⁶ potentially explaining why it is less efficiently recruited to H3K36me3 marked CGIs than DNMT3B.

Both points are related. I believe that additional experimental data or computational data that provide further insights into these questions will make the study a lot more significant.

We thank the reviewer for these useful comments. We believe the additional work to address them have strengthened the manuscript.

Minor point:

3. The sentence "Transcription deposition of H3K36me3 leads to DNMT3B-dependent methylation of CGIs in mouse ES cells and is associated with de novo methylation of imprinting control regions in mouse oocytes" is confusing and inaccurate, because genetic studies have shown that DNMT3A and its co-factor DNMT3L, but not DNMT3B, are essential for methylation of ICRs in germ cells.

This was a badly edited sentence in the original manuscript. We have revised it to the following:

Transcription induced deposition of H3K36me3 leads to Dnmt3b-dependent methylation of CGIs in mouse ES cells⁴¹. H3K36me3 is also associated with de novo methylation of imprinting control regions in mouse oocytes⁴². However, this is dependent on Dnmt3a and Dnmt3l but not Dnmt3b^{43,44}.

Reviewer #2 (Remarks to the Author):

As reviewer 2 summarises the key aspects of their comments at the end of their review, we have provided our responses in answer to these summarised points.

Key results:

In order to maintain fidelity of DNA methylation during mitosis DNMT and TET families of enzymes work synergistically. Maintenance DNA methylation is undertaken by DNMT1 and de novo DNA methylation by DNMT3a/3b. It is widely accepted that aberrant gain of DNA methylation at CpG islands occurs in colorectal cancer tumorigenesis, leading to altered gene expression of tumour suppressor and oncogenes. There is some evidence that CpG islands are targeted in a sequence dependant manner through de novo DNA methyltransferase activity. This paper sets out to determine where de novo DNA methyltransferases target CpG islands in colorectal tumourigenesis using ectopically manipulated in vitro cell culture of one colorectal cell line (HCT116) and normal epithelia. Conceptually this is a very relevant area to investigate in tumourigenesis as epigenetic targeting and manipulation are becoming therapeutic options. The authors suggest that their findings demonstrate a low level of de novo methylation at CpG islands, and that when it does occur it is mainly at CpG islands

which are marked by the histone modification H3K36me3 which is associated with transcriptional elongation. Furthermore it is suggested that as H3K36me3 marked CpG islands are highly methylated in normal colorectal epithelia and the HCT116 cell line, de novo DNA methylation in colorectal cancer tumorigenesis occurs at the same targets as normal colorectal tissue and is not altered in tumorigenesis.

Validity:

There are significant limitations in the approach that was used to test the experimental hypothesis. These limitations relate to the use of cell lines and the ability to translate these findings to in vivo systems and tissues.

1: Only one colorectal cell line was used for this work – when there are many available. The use of biological and technical replicates is not clear in the methods. There is one reference to biological replicates in the figure legend of figure 2e and a reference to technical replicates is made in the legend of Supplementary Figure 1. The authors need to clarify and include information regarding biological replicates.

2: Cell culture affects genome-wide DNA methylation – this has significant implications in the interpretation of the results presented in this paper and requires acknowledgement in the introduction and discussion. Rogers et al have described a lack of concordance between DNA methylation profiles of ependymoma cell lines and tissue (Oncotarget. 2018 Nov 23; 9(92): 36530–36541). This is particularly relevant when the authors conclude that their findings can be extrapolated to colorectal cancer tissue.

Originality and significance:

This is the first group to suggest these conclusions. However I have reservations regarding the ability to draw these conclusions from the work described alongside the limitations of using one colorectal cancer cell line and the effects of cell culture on genome-wide DNA methylation profiles.

I do, however appreciate the lack of alternative models available to test the experimental hypothesis. Zhang et al used an inducible transgenic mouse model focusing on the impact of genome-wide de novo DNMT3b activity across a number of cell types – with different findings to those presented in this paper (eLife. 2018; 7: e40757. Published online 2018 Nov 23. doi: 10.7554/eLife.40757 PMID: PMC6251628). This work is not referred to in the paper and is relevant.

Data & methodology:

Within the limitations described previously, the cell line work, piggyBac and experimental approach are appropriate and described with sufficient detail to allow another group to undertake these experiments.

The description of the data analysis undertaken requires more detail – for example “Infinium array data processing” (pg 21) requires information about whether this is

data from the Illumina HumMeth 27 or 450K arrays, what normalisation steps were taken to address batch effects etc.

Appropriate use of statistics:

Statistical tests and application are appropriate. However, description of error bars is missing in the legends of Figure 2d, 3b, 4b

Conclusions: Do you find that the conclusions and data interpretation are robust, valid and reliable?

Unfortunately I do not find the conclusions to be robust due to the limitations described previously regarding the use of one cancer cell line and the effects of genome-wide DNA methylation in cell culture.

Suggested improvements:

In order to have more certainty of the findings described in the paper I would suggest

1: Repeating the experiment with an alternative colorectal cancer cell line and normal colorectal tissue cell line.

We have now repeated key experiments in an additional colorectal cancer cell line, RKO. These experiments demonstrate that, similarly to HCT116 cells, ectopic copies of aberrantly methylated CGIs do not become methylated in RKO cells (included as Supplementary Fig. 1b, c). We also demonstrate enrichment of DNMT3B at H3K36me3 marked CGIs in RKO cells (included as Supplementary Fig. 3c,d).

During the period of revision, we were unable to obtain results from a normal colon tissue cell line as the lines we tested died after transfection. As transfection is an obligatory part of these experiments, this has prevented us generating results.

2: Reference to and discussion around the effect of cell culture on genome-wide DNA methylation and the impact this has on interpretation of these results.

As the reviewer states, cell culture is known to affect DNA methylation patterns. We have added analysis demonstrating that the aberrantly methylated CGIs we have tested are frequently methylated in colorectal cancer (Supplementary Fig. 1a) and showing that DNMT3B targets derived from our experiments in DKO cells are significantly enriched in CGIs marked by H3K36me3 in tumours:

DKO cell DNMT3B targets (from Fig. 2a) were also significantly enriched in colorectal tumour H3K36me3 marked CGIs ($p < 2.2 \times 10^{-16}$, Fisher's exact test) and depleted in colorectal tumour H3K4me3 marked CGIs ($p < 2.2 \times 10^{-16}$, Fisher's exact test).

We have also added discussion of the implications of the effects of cell culture on DNA methylation for our study:

A further potential limitation of the use of cell lines is that DNA methylation patterns are altered by cell culture⁵⁴. Little correspondence is reported between global methylation patterns in clinical ependymomas and cultured cell lines⁵⁵. However, we have previously shown that aberrantly methylated CGIs identified in breast cancer cell lines are also identified in clinical tumours⁵ and here we have focused on CGIs whose methylation is observed in vivo.

3: Clear indication of the use of biological and technical replicates and annotation of figure legends appropriately regarding error bars.

Biological and technical replicates:

Where replicates were done, they are noted in the figure legends. In the original manuscript this was limited to the cases the reviewer highlights (technical replication of mass spectrometry in the original Supplementary Fig. 1a and biological replication of the ChIP in the original Fig. 2e). We also performed the reintroduction of DNMT3B into DKO cells twice with similar results (original Supplementary Fig. 2e-g). However, as different promoters were used to express DNMT3B these are not strictly replicate experiments.

In the revised manuscript we have added additional replication of our results in another cell line as described above (Supplementary Fig. 1c,v and Supplementary Figs. 3b,c) and in two additional clinical cohorts as described below (Supplementary Fig. 5). Many of the new experiments were also biologically replicated and we have been careful to indicate this in the figure legends (Fig. 2d, Fig. 3, Supplementary Fig 2c, k and Supplementary Figure 3c, d).

Error bars:

We believe error bars were annotated in all figures. In the revised manuscript we have replaced these with data points where possible. The specific cases highlighted by the reviewer are now Fig. 2d, Fig. 4b and Fig. 5b. These are boxplots showing the distribution of data and are defined as follows in the relevant figure legends:

Lines=median; Box=25th-75th percentile; whiskers=1.5x interquartile range

4: Including additional detail to the data analysis methodology for the Illumina HumMeth Infinium array data.

As requested, we have provided additional detail in the methods specifying that the data were derived from Illumina Infinium 450k arrays. No batch correction was performed on the Illumina array data from TCGA. To demonstrate that batch effects do not explain our results, we now replicate our analysis in an independent dataset of colorectal tumours (from Fennell et al 2019, PMID: 30954552). In addition, we also observe similar results in a smaller third dataset containing adenomas in addition to colorectal tumours (see reviewer 3 comments, below). The fact that we derive similar results from the analysis of 3 independent datasets robustly demonstrates that batch effects do not underpin our findings in the clinical samples.

References:

The work of Zhang et al is not referenced and is relevant (eLife. 2018; 7: e40757. Published online 2018 Nov 23. doi: 10.7554/eLife.40757 PMID: PMC6251628).

We have added discussion of this study:

A recent study has shown that lowly-expressed H3K27me3 marked CGIs gain methylation in normal mouse tissues when high ectopic levels of the embryonic active form of Dnmt3b, Dnmt3b1, are expressed⁴⁸. However, the gains observed were not specific to the orthologues of those genes methylated in human tumours and the relative degree of Dnmt3b targeting of H3K36me3 marked loci was not assessed⁴⁸. The relevance of DNMT3B overexpression in human cancer has also been questioned and apparent upregulation is suggested to reflect the greater proportion of cycling cells in tumour tissues^{49,50}.

Clarity and context:

The abstract and introduction are clear.

The conclusion is written clearly, but for reasons described previously I do not feel that the findings can be extrapolated to colorectal cancer tissue.

We hope that the revisions we have provided provide additional demonstration of the relevance of our findings for tumours *in vivo*.

Reviewer #3 (Remarks to the Author):

This study uses the mismatch repair-deficient cell line HCT116 and its derivatives (DKO, DNMT3B-KO and DNMT1-KO) to investigate the mechanisms of CpG island (CGI) targeting by de novo DNMT activity. The findings disagree with prevailing models of sequence-specific (transcription factor-mediated) targeting, and propose a new, H3K36me3-based model, on the following evidence. First, results from a transposon-based approach for random, ectopic integration of selected CGIs into the HCT116 genome suggested that targeting of de novo DNMT activity is not sequence-specific. Second, introduction of DNMT3B into DKO cells revealed over 2,000 CGIs that gained methylation and the respective regions were marked by H3K36me3. Third, 5-aza-dC experiments showed that post-treatment remethylation was significantly faster at H3K36me3-marked CGIs compared to other CGIs. Fourth, re-analysis of TCGA-derived H3K36me3 ChIP-seq data from colorectal tumors and normal colon samples (including a set of matched normal-tumor pairs) showed that methylation levels were equally high in normal and tumor tissues, arguing against tumor-specific remodeling of H3K36me3 patterns at CGIs.

This work provides a comprehensive investigation into the mechanisms of de novo DNMT activity at CGIs in colorectal cancer. The observation that de novo DNMT activity is predominantly targeted at gene body CGIs marked by H3K36me3 is an important new finding supported by previous clues (references 17 and 36).

Comments:

1. As the authors state (Discussion, p. 10), the HCT116 colorectal cancer cell line models late stages of tumorigenesis after malignant transformation. Regarding the TCGA clinical colorectal tumor samples investigated, did they all represent carcinomas? Perhaps benign tumors (adenomas) were included as well?

The TCGA dataset analysed does not include adenomas. We have now undertaken an analysis of another dataset that does include adenoma samples (Luo et al 2014, Gastroenterology 147:418-29, doi: 10.1053/j.gastro.2014.04.039). This demonstrates that colorectal tumour H3K36me3-marked CGIs are also methylated in adenomas (data are included in Supplementary Fig. 5b).

2. Did the authors perform a closer analysis of the 2,238 CGIs that gained significant methylation in DKO cells upon DNMT3B introduction? How were the respective genes distributed according to different functional categories, for example?

We have performed an analysis of the GO-terms associated with the DNMT3B target CGIs in HCT116 cells that reveals a number of significantly enriched terms (included as Supplementary Fig. 2b). While significant, we observed very low enrichments for these terms suggesting the genes do not fall into known homogenous functional categories. We believe the target CGIs of DNMT3B are better understood by their location in gene bodies and marking by H3K36me3.

3. Discussion on p. 10 (“Our observations suggest that aberrant CGI hypermethylation could occur through an inefficient, slow process associated with low de novo DNMT activity”) and the right half of Fig. 5: If de novo DNMT activity is low, as observed by the authors, what is the (likely) mechanism behind aberrant tumor-specific methylation of CGIs based on available evidence?

We have added the following text to our discussion that clarifies the potential mechanisms responsible for the gain of methylation at aberrantly methylated CGIs:

The mechanism by which polycomb marked CGIs aberrantly gain methylation in cancer remains unclear². It has been proposed that TET dysfunction mediated by mutations or hypoxia underpins this epigenetic switch^{51,52}. Gains associated with TET-dysfunction could be expected to accumulate through the failure to remove aberrantly placed DNA methylation and thus could occur despite a lack of strong targeting of these CGIs by de novo DNMTs.

REVIEWERS' COMMENTS

Reviewer #1 (Remarks to the Author):

The issues raised by me have been satisfactorily addressed in the revised manuscript. One more suggestion I have is that, given the recent finding that the inactive DNMT3B3 isoform enhances DNMT3A/3B activities (Xu et al. Nature 2020; Zeng et al. Genes Dev 2020), the authors' suggestion that DNMT3B3 recruits DNMT3A to CpG islands marked by H3K36me3 may not be entirely correct. The possibility that DNMT3B3 stimulates DNMT3A activity should also be mentioned.

Taiping Chen

Reviewer #2 (Remarks to the Author):

I am happy with the new data and revised manuscript that the authors have provided.

Reviewer #3 (Remarks to the Author):

The authors have added important new data that strengthen their original conclusions of a H3K36me3-based mechanism of CGI targeting for methylation. Moreover, analysis of an independent set of benign and malignant colorectal tumors shows that the results obtained from cell line models apply to patient specimens. The revisions made are adequate and I have no additional concerns.

Response to reviewers round 2 NCOMMS-19-29594 Masalmeh *et al.*

We have addressed the remaining concerns of the reviewers in the manuscript and enclose a revised copy with changes highlighted in **blue**. In addition, this version addresses points raised by the editorial team.

Reviewers' comments (in *blue italics*)

Reviewer #1 (Remarks to the Author):

The issues raised by me have been satisfactorily addressed in the revised manuscript. One more suggestion I have is that, given the recent finding that the inactive DNMT3B3 isoform enhances DNMT3A/3B activities (Xu et al. Nature 2020; Zeng et al. Genes Dev 2020), the authors' suggestion that DNMT3B3 recruits DNMT3A to CpG islands marked by H3K36me3 may not be entirely correct. The possibility that DNMT3B3 stimulates DNMT3A activity should also be mentioned.

Taiping Chen

Reviewer #2 (Remarks to the Author):

I am happy with the new data and revised manuscript that the authors have provided.

Reviewer #3 (Remarks to the Author):

The authors have added important new data that strengthen their original conclusions of a H3K36me3-based mechanism of CGI targeting for methylation. Moreover, analysis of an independent set of benign and malignant colorectal tumors shows that the results obtained from cell line models apply to patient specimens. The revisions made are adequate and I have no additional concerns.

We thank the reviewers for their work in assessing the manuscript. As suggested by reviewer 1 we have included discussion to the two papers published since we submitted our revised manuscript (relevant sections highlighted in **bold**):

*We also observe gains of DNA methylation at H3K36me3 marked loci when catalytically inactive DNMT3B is introduced into DKO cells. DNMT3A levels are increased in DKO cells and DNMT3A and B can interact³⁰. A previous study suggested that catalytically inactive DNMT3B may recruit DNMT3A to H3K36me3 marked gene bodies by comparing the kinetics of remethylation in cells with and without DNMT3A⁴⁵. **The structure of the catalytic domains of DNMT3B3 and DNMT3A2 bound to a nucleosome has also been solved⁴⁶. These observations could also be explained by model where DNMT3A constitutively localises to H3K36me3 but that DNMT3A-DNMT3B hetero-complexes have higher catalytic activity³⁰. This model is supported by a recent study showing that Dnmt3b3 can act as an accessory protein for Dnmt3a stimulating its catalytic activity at repetitive sequences⁴⁷. Here, we directly confirm that DNMT3A is also more efficiently recruited to H3K36me3 marked CGIs in the presence of DNMT3B.***